# Bacteriophage defends murine gut from *Escherichia coli* invasion via mucosal adherence

Jiaoling Wu[1], Kailai Fu[1], Chenglin Hou[1], Yuxin Wang[1], Chengyuan Ji[1], Feng Xue[1], Jianluan Ren[1], Jianjun Dai [1,2] ✉, Jeremy J. Barr [3] ✉ & Fang Tang[1] ✉

Bacteriophage are sophisticated cellular parasites that can not only parasitize bacteria but are increasingly recognized for their direct interactions with mammalian hosts. Phage adherence to mucus is known to mediate enhanced antimicrobial effects in vitro. However, little is known about the therapeutic efficacy of mucus-adherent phages in vivo. Here, using a combination of in vitro gastrointestinal cell lines, a gut-on-a-chip microfluidic model, and an in vivo murine gut model, we demonstrated that a *E. coli* phage, øPNJ-6, provided enhanced gastrointestinal persistence and antimicrobial effects. øPNJ-6 bound fucose residues, of the gut secreted glycoprotein MUC2, through domain 1 of its Hoc protein, which led to increased intestinal mucus production that was suggestive of a positive feedback loop mediated by the mucus-adherent phage. These findings extend the Bacteriophage Adherence to Mucus model into phage therapy, demonstrating that øPNJ-6 displays enhanced persistence within the murine gut, leading to targeted depletion of intestinal pathogenic bacteria.

Bacteriophage (phage) are viruses that specifically infect bacteria and are the most abundant entities in the biosphere[1–3]. Within the mammalian gut, the abundance of phages is estimated to be closer to a 1:1 ratio with their bacterial hosts, and they both contribute to human health[4,5]. Due to the increase in multidrug resistance, particularly within disease-causing bacteria, there has been a growing interest in the use of phages to treat bacterial infections[6,7]. Although phage therapy has shown promise in selectively targeting virulent bacteria in the gut without harming symbiotic bacteria, there is still uncertainty regarding its overall efficacy and practical application in the gastrointestinal tract[5,8]. Recent examples include the use of a phage cocktail targeting *Klebsiella pneumoniae* that specifically killed harmful bacteria in the intestinal tract of mice. This cocktail was subsequently given to human patients, with subjects having no adverse reaction to phage administration and high concentrations of phage remaining in the feces after cessation of administration[6]. Similarly, the

administration of phages that targeted *Enterococcus faecalis* strains resulted in reduced intestinal colonization and inflammation in humanized mice[9]. Conversely, a study using T4 phage to treat diarrhea in infants reported no clinical benefits, citing the reason for treatment failure being the inability of the phages to replicate within the intestine[10]. The targeted bactericidal effect of phages is often diminished within the intestinal environment due to the extreme microbial diversity, rapid changes in phage concentration, and the inability of phages to find a susceptible host and effectively multiply within this locale[11]. In addition, phages may directly interact with and be internalized by the mammalian cells, while others may not be well-adapted to the gastrointestinal environment[12,13]. In summary, the mammalian intestine presents challenges for phage therapy, necessitating further research to fully leverage phages' potential against enteric pathogens.

The human gastrointestinal mucosal surface is considered the primary barrier of infection against various pathogens, including

[1]College of Veterinary Medicine, Nanjing Agricultural University; Key Laboratory of Animal Bacteriology, Ministry of Agriculture, Nanjing, China. [2]School of Pharmacy, China Pharmaceutical University; Engineering Research Center for Anti-infective Drug Discovery, Ministry of Education (ERCADD), Nanjing, China. [3]School of Biological Sciences, Monash University, Victoria, Australia. ✉e-mail: jjdai@njau.edu.cn; jeremy.barr@monash.edu; tfalice@126.com

bacteria, fungi, and viruses[14–16]. The mucosal epithelium consists of goblet cells, M cells, enterocytes, and enteroendocrine cells, which are covered by a structured mucosal layer[17–19]. Mucus layers are present throughout the body and mainly consist of host-secreted mucin glycoproteins[20,21]. These mucins are large, highly glycosylated proteins that are expressed predominantly by goblet cells and are encoded by the MUC gene family, which consists of 20 known proteins, with Mucin-2 (MUC2) being the primary variant expressed within the gut[20,22]. MUC2 is glycosylated by both O- and N-linked glycans that protect the epithelium from microbial invasion and digestive enzymes[17,22,23]. Mature mucin glycoproteins are then terminally decorated with sialic acid or fucose residues, which are likely a food source for bacteria in the intestine[17,24,25].

The gut mucus environment is conducive to the enrichment and localization of both bacteria and phage[26–28]. Previous work proposed the Bacteriophage Adherence to Mucus (BAM) model whereby certain phage types adhere to mucins through binding interactions between capsid-displayed Immunoglobulin-like (Ig-like) domains and mucin glycan residues[29–31]. While the BAM model has been demonstrated to impact bacterial colonization in vitro, its efficacy within the mammalian gut remains unexplored. Gabriel et al. reported that the phage-mucus interaction provided protection for rainbow trout against *Flavobacterium columnare* infection[32]. Another study demonstrated that phages were able to evolve in response to mammalian-derived mucosal environments within a gut-on-a-chip microfluidic device, resulting in alterations in the phages' binding affinity to mucin glycans[29]. In contrast, the ecology and persistence of crass-like phages were suggested to be more in line with the Piggyback-the-Winner model rather than the BAM model in vivo, although it should be noted these two models are not mutually exclusive[33].

In this work, we aimed to investigate the efficacy of mucus-adherent phages to reduce the colonization of mucosal enteric pathogens within a phage-therapy model of the murine intestine. We used the phage, øPNJ-6, which was isolated from chicken feces and displayed a Hoc protein that was predicted to facilitate its adherence to mucus. Using in vitro gastrointestinal cell lines, a gut-on-a-chip microfluidic model, and in vivo murine model, we show that mucus-adherent phages are effective agents for enteropathogen decolonization of the gut.

**Table 1 | Homology analysis of Hoc amino acid sequences of strain øPNJ-6 with other strains**

| Host | Strains | GenBank accession no. | Hoc identity (%) | Query cover (%) |
|------|---------|-----------------------|------------------|-----------------|
| *E.coli* | T4 | NP_049793.1 | 87.8 | 100 |
| *E.coli* | ST0 | YP_009608485.1 | 97.9 | 100 |
| *E.coli* | vB_EcoM-ZQ3 | QWY13362.1 | 93.9 | 100 |
| *E.coli* | RB3 | YP_009098563.1 | 93.4 | 100 |
| *E.coli* | RB32 | YP_803120.1 | 92.0 | 100 |
| *E.coli* | vB_EcoM_ESCO47 | UPW39467.1 | 92.0 | 100 |
| *E.coli* | YP_010247135.1 | HX01 | 93.6 | 100 |
| *E.coli* | ECO07P1 | WAX12894.1 | 92.6 | 100 |
| *Shigella* | phi25-307 | YP_010093233.1 | 95.7 | 100 |
| *Shigella* | CT01 | UDY80572.1 | 93.6 | 100 |
| *Shigella* | ESh15 | URY11297.1 | 94.7 | 100 |
| *Yersinia* | fPS-2 | YP_010077062.1 | 92.6 | 100 |
| *Yersinia* | PYPS2T | YP_010077170.1 | 89.9 | 100 |
| *Yersinia* | vB_YepM_ZN18 | YP_010077679.1 | 89.4 | 100 |
| *Salmonella* | pSe_SNUABM_01 | YP_010075447.1 | 92.0 | 100 |
| *Salmonella* | Lv5cm | QVW08985.1 | 91.8 | 100 |
| *Salmonella* | GRNsp7 | USW07282.1 | 92.0 | 100 |

# Results

## øPNJ-6 is a mucus-adherent phage that kills *Enterotoxigenic Escherichia coli*

To reduce the use of antibiotics and alleviate the development of antibiotic resistance, we set out to use phage therapy to combat *Enterotoxigenic Escherichia coli* (ETEC) infections. We isolated a T4-like phage, designated as øPNJ-6, from chicken feces, which demonstrated strong lytic activity against *E. coli* SH232. This strain was isolated from the diarrheal feces of piglets and encoded virulence genes *STa* and *K99* (Supplementary Fig. 1a) and was therefore identified as ETEC. We performed whole genome sequencing and determined øPNJ-6 genome was 170.4 Kbp in size and contained 273 Open Reading Frames (ORFs). Notably, øPNJ-6 (GenBank: OQ076693.1) encodes a highly antigenic outer capsid protein (Hoc, 1131 bp) that shares 87.8% homology with that of T4 phage Hoc in terms of amino acids sequence. We found that Hoc-like proteins with high homology exist in many phages targeting various *Enterobacteriaceae* bacteria (e.g. *Shigella*, *Yersinia*, and *Salmonella*) (Table 1). Previous studies reported that the Hoc protein of phage T4 interacts with the glycan components of mucin, leading to the enrichment of the phage in mucus[29,30]. First, we investigated whether this phenomenon occurs in øPNJ-6. Using an in vitro assay, we coated agar plates with 1% mucin and demonstrated that øPNJ-6 adhered to mucin-coated plates at a significantly higher titer than plates without mucin (Supplementary Fig. 1b). Additionally, we found that the propagating phage titer was 10 times higher in LB liquid media that contained 1% mucin than compared with normal LB after 24 h incubation (Supplementary Fig. 1c), indicating that mucins seem to stabilize phage against decay in vitro[34]. Hence, these results suggested that mucin played a pivotal role in facilitating enhanced phage enrichment and replication in vitro.

## øPNJ-6 protects in vitro HT-29 colonic epithelial cells from ETEC infection

Next, we aimed to investigate whether øPNJ-6 could adhere to colonic epithelial cells (HT-29) cultured in vitro, which are known to produce gastric mucins, and provide protection against bacterial colonization. In our preliminary observations, we noted that HT-29 gastro-epithelial cells cultured in vitro exhibited a measurable but weak secretion of MUC2, while Madin-Darby Bovine Kidney (MDBK) cells as control did not demonstrate such secretion (Supplementary Fig. 1d). Additionally, we assessed the potential cytotoxicity of øPNJ-6 lysate on the cells and found no significant detrimental effects (Supplementary Fig. 1e). To investigate the tripartite interactions and protective effects, we pretreated HT-29 cells with øPNJ-6 for 30 min, followed by the introduction of its bacterial host ETEC SH232. In the phage-treated groups, the concentration of attached ETEC cells was significantly lower than that of the non-phage-treated groups at 3 h, 6 h, 12 h and 24 h (Fig. 1a). The viability of HT-29 cells was significantly higher in phage-pretreated groups than that of non-phage-treated groups at 3 h ($P < 0.01$), 6 h ($P < 0.0001$), 12 h ($P < 0.001$) (Fig. 1b), suggesting that phage prevented ETEC from mediating infection in vitro. The concentration of ETEC SH232 in the supernatant in phage-treated groups was also significantly lower than that of the non-phage-pretreated groups (Supplementary Fig. 1f), as numerous phages existed in the supernatant (Supplementary Fig. 1g). This phenomenon was verified by epifluorescence microscopy observations showing the marked reduction of ETEC upon phage pretreatment (Fig. 1c).

To better simulate intestinal-like micro-environments, we performed the above experiments using a gut-on-a-chip microfluidic model, which was fabricated as previously described (Supplementary Fig. 2a)[11,31]. HT-29 cells were seeded on a glass coverslip within the gut-on-a-chip microfluidic devices until a confluent epithelium was established, followed by perfusion with media with or without phage, then inoculation with bacteria ETEC SH232. Phage pretreatment

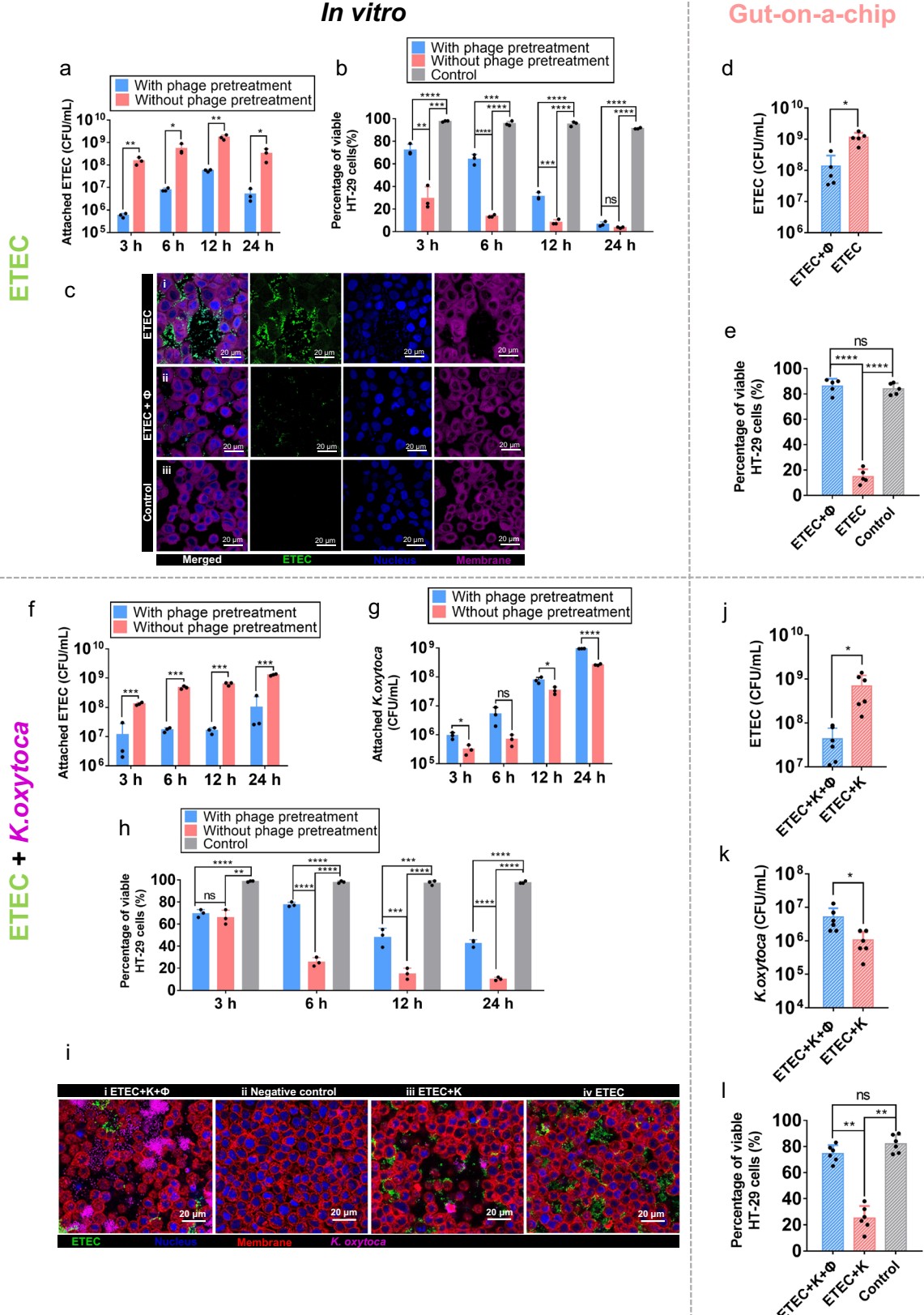

significantly reduced the number of adherent ETEC SH232 within the gut-on-a-chip (Fig. 1d) compared with non-pretreatment group, which also correlated with an increase in cell viability (Fig. 1e). Importantly, when utilizing this gut-like microfluidic model, cells in phage pretreated were almost completely protected from bacterial infection at 24 h (Fig. 1e), while in the above in vitro model, cells were mostly dead at 24 h even in phage pretreated group (Fig. 1b). Additionally, more than $10^9$ PFU/mL of øPNJ-6 remained attached to cells within the gut-on-a-chip under flow conditions at 24 h (Supplementary Fig. 2b). These data demonstrated that øPNJ-6 prevented host bacterial cells from mediating infection at the gut-like microfluidic model.

**Fig. 1 | Phage øPNJ-6 protects HT-29 cells from ETEC invasion.** In vitro model with ETEC: the number of ETEC attached to HT-29 cells (**a**), the proportion of live HT-29 cells in the phage pre-treatment or phage-free group (**b**), at different time points (3 h, 6 h, 12 h, and 24 h). **c** Fluorescence microscope photograph of HT-29 cells with different treatments in vitro. Green indicates ETEC, red indicates the cell membrane and blue indicates the nucleus. (**c-i**) ETEC was incubated with HT-29 cells for 3 h without prior phage treatment (×100); (**c-ii**) HT-29 cells were pre-treated with øPNJ-6, followed by incubated with ETEC for 3 h (×100); (**c-iii**) HT-29 cells were cultured without ETEC or øPNJ-6 treatment (×100). Scale bars, 20 μm. Gut-on-a-chip model with ETEC: (**d**) the number of ETEC in the phage-pretreated and phage-free treatment groups; (**e**) the cell survival in the presence or absence of phage pre-treatment. In vitro model with ETEC and *K.oxytoca*: the number of ETEC (**f**) and *K.oxytoca* (**g**) adhering to HT-29 cells, and the proportion of live HT-29 cells in the phage pre-treatment or phage-free group (**h**), at different time points (3 h, 6 h, 12 h, and 24 h). **i** Fluorescence microscope photograph of the competitive assay. Green indicates ETEC, purple represents *K.oxytoca*, red indicates the cell membrane and blue indicates nucleus; (**i-i**) ETEC and *K. oxytoca* were added to the cells and incubated for 3 h after phage treatment (× 63); (**i-ii**) HT-29 cells were not infected with any phage or bacteria (× 63); (**i-iii**) ETEC and *K. oxytoca* were added to the HT-29 cells and incubated for 3 h without phage (× 63); (**i-iv**) HT-29 cells were incubated with ETEC for 3 h (×63). Scale bars, 20 μm. Gut-on-a-chip model with ETEC and *K.oxytoca*: the number of ETEC (**j**), *K.oxytoca* (**k**), and the proportion of live HT-29 cells (**l**) in the gut-on-a-chip at 24 h. Data are presented as mean values ± SD (*n* = 3 biologically independent experiments) and *P*-values are calculated by Multiple *t* test one per row (*$P \leq 0.05$; **$P \leq 0.01$; ***$P \leq 0.001$; ****$P \leq 0.0001$). Source data are provided as a Source Data file.

Both commensal and pathogenic bacteria exist in a state of competition for both space and nutrients on the intestinal mucosal surface[5,17,35,36]. To better replicate the microbial community within the intestine, we utilized the intestinal symbiotic bacterium *Klebsiella oxytoca* (*K. oxytoca*), to assess the adhesion and competition capacities of ETEC following phage pretreatment. Firstly, using the in vitro static tissue culture model, both ETEC and *K. oxytoca* were added to HT-29 cells after pretreatment with øPNJ-6. As shown in the Fig. 1f, g, øPNJ-6 effectively inhibited the propagation of its host bacterium ETEC without impacting the growth of *K. oxytoca*, with similar trends observed in the supernatants (Supplementary Fig. 2c, d). In the presence of commensal strain *K. oxytoca* and pathogenic strain ETEC, phage øPNJ-6 adhered to HT-29 cells (Supplementary Fig. 2e) and resulted in the protection of cells at later time points (Fig. 1h). We observed similar results under epifluorescence microscopy, where øPNJ-6 was found to specifically lyse ETEC strains, without affecting the commensal bacteria *K. oxytoca* (Fig. 1i). Comparatively, in the gut-on-a-chip model we observed that pre-treatment with øPNJ-6 led to a more pronounced reduction in ETEC (Fig. 1j) compared to prior experiments without *K. oxytoca* (Fig. 1d), highlighting the importance of microbial ecology within the intestine. With and without *K.oxytoca* in gut-on-a-chip devices, øPNJ-6 exhibited strong mucus-adherence (Supplementary Fig. 2b). With øPNJ-6 specifically targeting ETEC, we observed that *K. oxytoca* grew to higher abundance in phage pretreatment group than that of the non-phage pretreated group (Fig. 1k), and HT-29 cells had increased viability in the phage pretreatment group (Fig. 1l), suggesting a synergistic effect between øPNJ-6 and commensal gut microbes in defending against ETEC infections.

## Mucus enhances the adherence and antimicrobial properties of øPNJ-6 in the gastrointestinal tract

Previous studies have indicated that the enrichment of T4 phage within the mucosa is dependent on the presence of mucins rather than other macromolecular components[30,31]. Mucus gel layers at the mucosal surface are primarily composed of mucins, which are predominantly secreted by epithelial goblet cells. To determine whether mucins play a role in promoting bacterial killing by this phage øPNJ-6, we utilized the mucolytic agent, *N*-acetyl cysteine (NAC), to remove mucins secreted by HT-29 cells. NAC was added to the cells for a 30-min incubation period to reduce the mucin glycoprotein, which is primarily composed of MUC2. Subsequently, we measured the MUC2 content following NAC pretreatment and observed that there was a significant reduction in MUC2 when the concentration of NAC was 15 mM (Supplementary Fig. 2f), with only a slightly adverse effect on cell viability (Supplementary Fig. 2g).

We then assessed the potential impact of NAC pretreatment on the number of ETEC, cell viability, and phage attachment capacity under static in vitro conditions. We speculated that if mucus is removed by NAC, the number of phages adhered to cells will significantly decrease (Supplementary Fig. 2h, i). However, our results demonstrated that the number of øPNJ-6 attachments did not decrease following NAC pretreatment (Fig. 2a), and øPNJ-6 maintained its efficacy in killing the target bacterium ETEC SH232 in vitro (Fig. 2b and Supplementary Fig. 2i). There was also no significant change in the number of free phage with or without NAC pretreatment (Supplementary Fig. 2j). These findings indicated that, even in the presence of NAC-reduced mucus layer, ETEC remained susceptible to lysis by øPNJ-6 in vitro.

Interestingly, we discovered a starkly contrasting outcome when we conducted the above experiment with HT-29 cells in gut-on-a-chip microfluidic devices that simulate gut-like conditions. In the gut-on-a-chip model, HT-29 cells were cultured with media containing NAC (15 mM) or without NAC for 50 min at a flow rate of 120 μL/h, followed by the addition of phage and ETEC. The results revealed a 100-fold decrease in the number of phage in the NAC-pretreated group (Fig. 2c), leading to a significant decrease in bacterial killing (Fig. 2d) and cell viability (Fig. 2e).

Inconsistent patterns in bacterial abundance and levels of phage adhered to the mucus between in vitro and gut-on-a-chip models were observed. We speculated that the presence of bacteria and the associated phage propagation may affect our quantification of mucus-adherent phages. Thus, we performed the above experiments in the absence of bacteria. Using our in vitro model, øPNJ-6 was incubated with HT-29 cells that had undergone pretreatment with NAC, resulting in a significant but weak reduction in the number of øPNJ-6 compared to cells without NAC pretreatment (Supplementary Fig. 2k). These findings suggested that the adherence ability of the øPNJ-6 to HT-29 cells was diminished when the mucus layer was reduced through NAC pretreatment. Next, in the gut-on-a-chip model, øPNJ-6 was incubated with HT-29 cells in the absence of ETEC, with a 30-min pause in flow rate to allow resident phages time to adhere to the mucosal surface. When combined with NAC pretreatment, we observed a significant reduction in the number of phages adhering to cells on the gut-on-a-chip (Supplementary Fig. 2l). This data demonstrates that NAC pretreatment reduced the ability of øPNJ-6 to attach to the cell surface due to the removal of mucus.

Many phenomena observed in vitro fail to replicate when tested in vivo, thus necessitating the confirmation of in vitro experimental results through in vivo tests to substantiate their value. To further extend the relevance of øPNJ-6 to the gut, we evaluated phage adherence using an in vivo murine model. We administered øPNJ-6 via oral gavage to mice that were pretreated with NAC for 10 days to reduce gastrointestinal mucus, followed by øPNJ-6 oral gavage. Mice without NAC treatment served as a control. Seven hours later, the small intestine, cecum, and colon of the mice were dissected, homogenized, and centrifuged to remove pellets. The supernatant was collected, and phage was quantified via double-layer agar plates. The results showed that NAC-pretreated mice had a 2- to 3-fold reduction in phage attachment in the small intestine

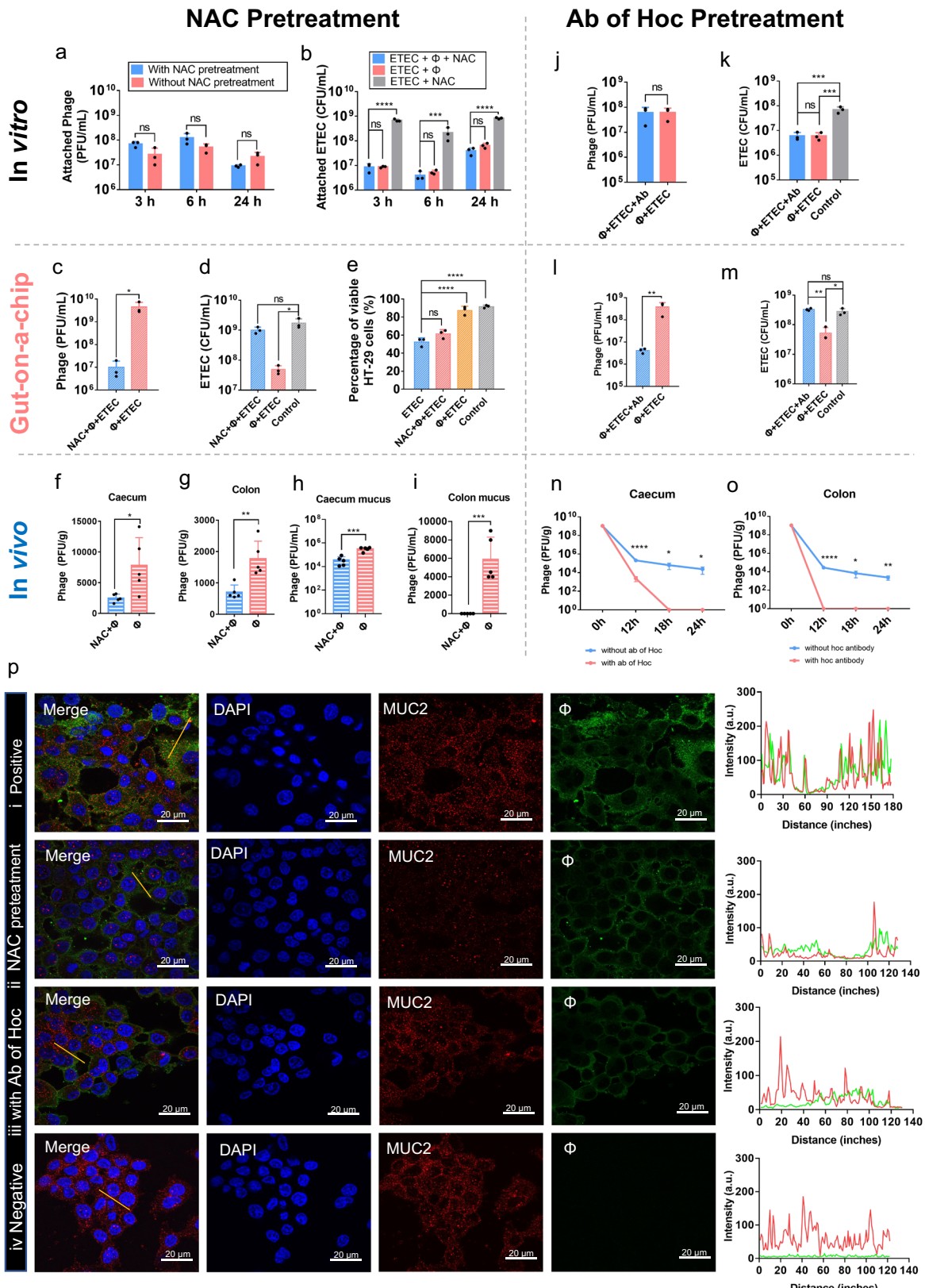

(Supplementary Fig. 2m), cecum (Fig. 2f), and colon (Fig. 2g) compared to the non-NAC pretreatment controls, but this change is relatively weak. To further investigate the phage affinity for the mucus layer in vivo, we performed the above experiments by removing lumen contents, collecting the mucus layer of the cecum and colon, and counting the adhered phage. We observed a significant decrease in øPNJ-6 adhered to the mucus layer of the caecum (Fig. 2h) and colon (Fig. 2i) of NAC pretreated mice compared to non-pretreated groups. Our findings suggested that reduction of the mucus layer through NAC pretreatment diminished the capacity of øPNJ-6 to adhere to the gastrointestinal tract and mediate its antimicrobial effects.

**Fig. 2 | Mucin and Hoc were involved in the interaction between øPNJ-6 and the intestinal tract.** NAC pre-treatment: The number of phage (**a**) and ETEC (**b**) attached to HT-29 cells with or without NAC or phage pre-treatment at different time points (3 h, 6 h, 24 h) in vitro. The number of phage (**c**) and ETEC (**d**) attached to HT-29 cells, and the proportion of live HT-29 cells (**e**) pre-treated with or without NAC or phage pre-treatment at 24 h in the gut-on-a-chip. The number of phage adhering to the caecum (**f**), colon (**g**), or the mucus layer of the caecum (**h**) and colon (**i**) of mice pre-treated with or without NAC in the mouse model. Hoc antibody treatment: the number of phage (**j**) and ETEC (**k**) attached to HT-29 cells pretreated with phage or Hoc antibody blocked phage at 3 h in vitro. The number of phage (**l**) and ETEC (**m**) attached to HT-29 cells pre-treated with phage or Hoc antibody blocked phage at 24 h in the gut-on-a-chip. The number of phage adhering

to the caecum (**n**) and colon (**o**) of mice pre-treated with phage or Hoc antibody blocked phage. **p** Fluorescence microscope photograph of the binding of øPNJ-6 to MUC2. (**p**-i) øPNJ-6 co-localized with MUC2 in the positive control group (×100); (**p**-ii) The number of øPNJ-6 adhering to MUC2 was decreased after NAC pretreatment (×100); (**p**-iii) The number of øPNJ-6 adhering to MUC2 was diminished after Hoc antibody blocking (×100); (**p**-iv) HT-29 cells in their natural state shown no presence of øPNJ-6 (×100). Red indicates MUC2, green indicates øPNJ-6, and blue indicates cell nucleus. Scale bars, 20 µm. Data are presented as mean values ± SD ($n$ = 3 independent experiments) and two-tailed $P$-values are calculated Multiple $t$ test one per-row (*$P \le 0.05$; **$P \le 0.01$; ***$P \le 0.001$; ****$P \le 0.0001$). Source data are provided as a Source Data file.

## Phage øPNJ-6 adheres to the gastrointestinal tract via the Hoc protein

Previous research has indicated that Ig-like domains displayed via structural proteins of phage T4 play a crucial role in the adherence of phage to mucus in vitro[30,37]. Guided by this finding, we postulated that a similar Hoc protein may also augment the attachment of øPNJ-6 in the gastrointestinal tract (Supplementary Fig. 2h-ii). To test this hypothesis, we carried out the following experiments. First, we expressed the Hoc protein of øPNJ-6 (Supplementary Fig. 3a) and generated a polyclonal antibody against it (Supplementary Fig. 3b). Next, we treated øPNJ-6 with its anti-Hoc polyclonal antibody and determined that this did not affect the ability of the phage infect its host bacteria ETEC in vitro (Supplementary Fig. 3c). We conducted Hoc antibody blocking assays in vitro to evaluate on the phage's adherence to HT-29 cells. øPNJ-6 ($10^8$ PFU/mL) was incubated with the Hoc polyclonal antibody at 37 °C for 30 min and then added to HT-29 cells for a 30 min pretreatment before the addition of ETEC SH232 ($10^7$ CFU/mL). Results showed that there was no significant difference in the levels of phage adhesion (Fig. 2j), ETEC adhesion (Fig. 2k), or cell survival (Supplementary Fig. 3d) between the blocked and non-blocked groups within in vitro experiments. However, when we performed the same experiments within gut-on-a-chip microfluidic devices, the number of phage attachments was noticeably reduced upon polyclonal antibody pretreatment (Fig. 2l), resulting in a significant decrease in bacterial killing (Fig. 2m) and cell viability (Supplementary Fig. 3e). Considering the inconsistent trend in the number of adherent phage in vitro static tissue culture and gut-on-a-chip microfluidic models, we designed additional bacteria-deficient experiments. In the absence of ETEC SH232, HT-29 cells were incubated with either free phage or phage pre-blocked with Hoc antibody. We observed a reduction in the number of phage attached to the cells after Hoc blocking, both in vitro (Supplementary Fig. 3f) and in gut-on-a-chip microfluidic devices (Supplementary Fig. 3g), indicating that Hoc played a crucial role in phage adhering to mucus.

Next, an immunofluorescence assay (IFA) was conducted to examine the interaction between øPNJ-6 and the mucosal layer in the gut-on-a-chip model. The experimental images revealed a clear co-localization of øPNJ-6 and MUC2 within the gut-on-a-chip system (Fig. 2p-i). However, when NAC was administered or when Hoc was blocked, the co-localization between øPNJ-6 and MUC2 was significantly reduced or eliminated (Fig. 2p-ii, iii), whereas no fluorescent green coloration was observed in the negative control (Fig. 2p-iv).

To control for the impact of flow dynamics on our chip-based assays, we adopted a sophisticated 'pump-paused' protocol, inspired by the methodology outlined in Tovaglieri et al.[38] This approach involved halting the fluidic pump for 30 min following the introduction of øPNJ-6 and ETEC into the gut-on-a-chip system, aligning with the incubation period used in our static in vitro assays. Results from this modified setup indicated that the free phage maintained a significant advantage in reducing ETEC numbers over those pretreated with Hoc-blocking antibodies or NAC (Supplementary Fig. 3h, i). Specifically, when using the 'pump-paused' protocol, the cell viability of

the cohorts receiving either antibody or NAC pretreatment showed significantly higher levels of cell death when compared to the phage pretreatment group (Supplementary Fig. 3j). These findings suggest that the Hoc protein plays a crucial role in the interaction between the phage and HT-29 cells within the gut-on-a-chip microfluidic model. However, under in vitro conditions where there is no fluid flow, stagnant conditions facilitate the propagation of both øPNJ-6 and ETEC in the supernatant where the effect of Hoc and its antibody blocking is marginal. Comparatively, within the gut-on-a-chip microfluidic device, continual fluid flow across the mucosal surface facilitates the removal of microorganisms in the supernatant and amplifies any mucosal interactions. Here, we observed significant impacts of antibody-mediated Hoc blocking, as seen through reduced phage adherence, increased bacterial numbers, and decreased cell survival. These results highlight the utility and relevance of gut-on-a-chip microfluidic devices over static in vitro tissue culture to capture these impacts.

To determine the influence of Hoc protein on the attachment of øPNJ-6 in the real intestinal tract, we conducted experiments using a mouse model. Four-week-old mice were gavaged with free phage or Hoc antibody-blocked phage øPNJ-6 in the absence of the ETEC host. The blocked phage was incubated with Hoc polyclonal antibody for 1 h at 37 °C. Our results revealed that øPNJ-6 was present in the gut of the mice for more than 24 h, even without the presence of susceptible bacteria (Figs. 2n, o). However, upon antibody pretreatment number of phage persisting in the intestinal tract showed more than 2-log decrease at 12–24 h in the cecum (Fig. 2n) and the colon (Fig. 2o). An examination of feces at 24 h revealed high levels of phage across all samples that were not pretreated with the specific Hoc antibody, with a concentration of $10^5$ PFU/g, while phage could not be detected in feces at 12 h in the antibody-coated group (Supplementary Fig. 3k). These findings suggested that Hoc protein plays a crucial role in promoting phage adherence and persistence in the murine gut. The obtained results suggested that the binding of øPNJ-6 to mucosal cells is due to the interaction between Hoc and the mucus layer.

## Hoc protein of øPNJ-6 can bind to MUC2

We established that the Hoc protein plays a crucial role in facilitating the enrichment of øPNJ-6 in mucus. Mucus is a complex biomaterial that covers the epithelial surfaces throughout the body, and its composition is largely determined by the presence of mucins, a family of 20 MUC genes. Of these genes, the most abundant component in the gut is MUC2, which is a glycoprotein that undergoes O- and N-glycosylation modifications[39–41]. Given the abundance of MUC2 in the mucus and its potential for interaction with phage, we hypothesized that there may be a relationship between øPNJ-6 Hoc and mammalian MUC2. To test this hypothesis, co-immunoprecipitation (CO-IP) experiments were performed between Hoc and MUC2. We utilized both HT-29 and LS174T cells to obtain MUC2, as previous study have indicated that the LS174T cell line possesses a higher concentration of mucin compared with the HT-29 cell line[42]. Our results demonstrated that the Hoc protein could bind with MUC2 secreted by HT-29 (Fig. 3a) and LS174T (Fig. 3b).

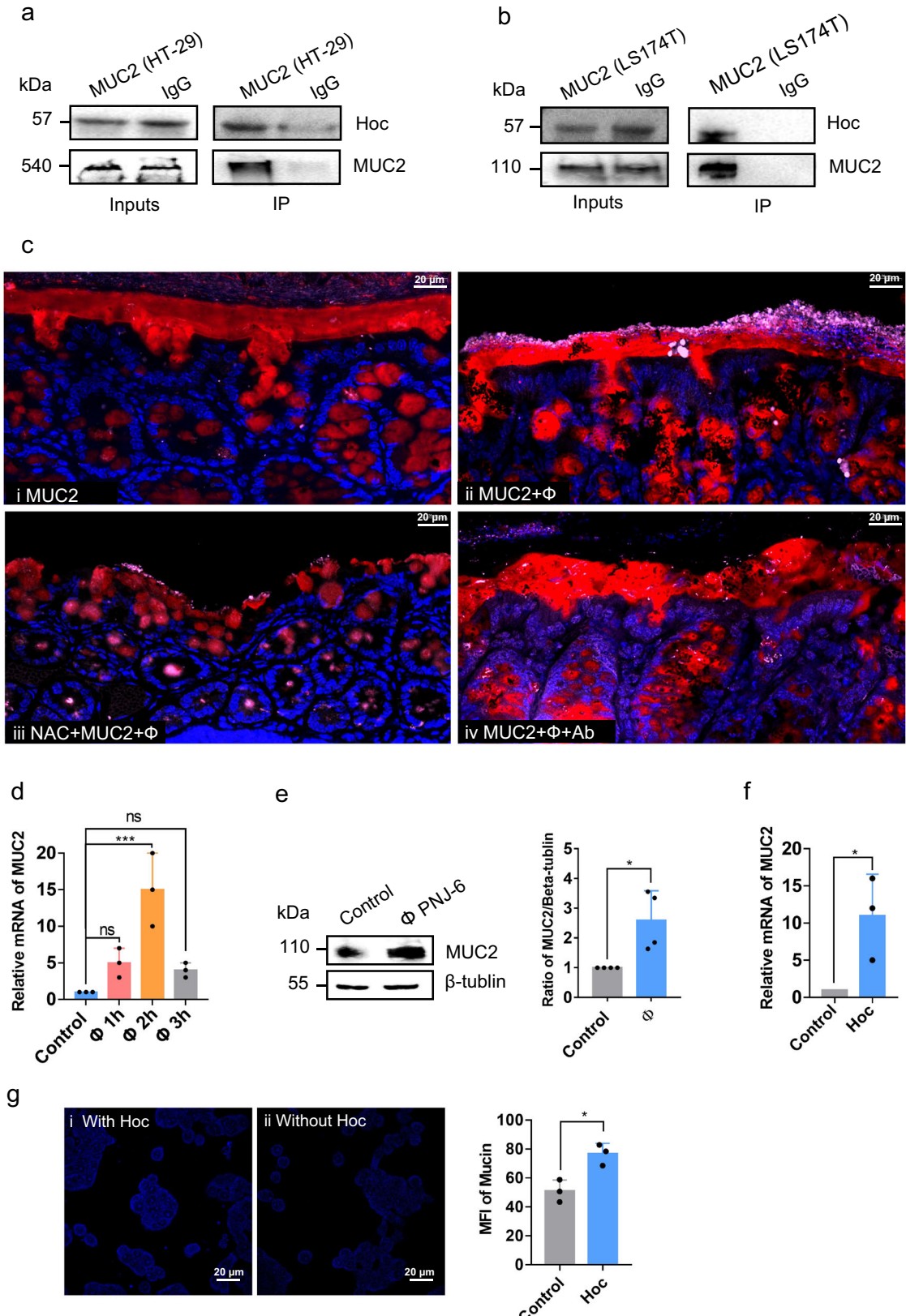

Next, we investigate whether øPNJ-6 could bind the MUC2 glycoprotein in vivo. To determine this, we observed the adhesion of phages to mouse colon under different conditions. Mice were either gavaged with Hoc-antibody-blocked øPNJ-6, free phage øPNJ-6, or gavaged with NAC to reduce the mucus before application with øPNJ-6. Our results showed that MUC2 was abundant on the epithelial surface

of the mouse colon (Fig. 3c-i), providing targets for phage adherence (Fig. 3c-ii). When mice were pretreated with NAC, we observed a reduction of colonic MUC2, resulting in a reduction in the number of adherent phages (Fig. 3c-iii). Similarly, when the capsid protein Hoc of øPNJ-6 was blocked by antibodies, we observed a reduction in the number of phages that were unable to effectively adhere to the mucus

**Fig. 3 | Phage øPNJ-6 promotes mucin secretion from HT-29 cells.** Immuno-precipitation of Hoc protein interacting with MUC2 obtained from HT-29 cell lysates (**a**) or LS174T cells (**b**). **c** Immunofluorescence photograph of a section of mouse colon. The red color represents MUC2, blue DAPI indicates cell nucleus, and pink indicates phage øPNJ-6; (**c**-i) Normal mouse colon without any treatment (×60); (**c**-ii) Abundant phage adhered to the colon in mice without NAC treatment (×60); (**c**-iii) After NAC treatment, MUC2 in the mouse colon was significantly reduced, leading to a sharp decrease of phage adhering to the colon (×60); (**c**-iv) Phage was blocked by Hoc antibody, resulting in a significant decrease of phage adhering to the colon (×60). Scale bars, 20 μm. **d** mRNA transcriptional level of MUC2 in HT-29 cells when incubated with øPNJ-6 for 1, 2, 3 h. **e** Expression level of MUC2 in HT-29 cells when incubated with øPNJ-6 for 2 h; The bar represents gray value analysis of WB. **f** mRNA transcriptional level of MUC2 in HT-29 cells when incubated with Hoc protein for 2 h; (**g**) Fluorescence microscopy photograph of mucus expression in HT-29 cells was incubated with or without Hoc protein for 2 h. The blue color indicates mucus (×60); The bar represents Mean Fluorescence Intensity (MFI) of mucin; Data are presented as mean values ± SD $P$-values are calculated by One-way ANOVA (**d**) or unpair Student's $t$ test (**e**–**g**) ($n = 3$ biologically independent experiments). To indicate significance, one symbol $p < 0.05$, two symbols $p < 0.01$, three symbols $p < 0.001$, four symbols $p < 0.0001$. Source data are provided as a Source Data file.

layer of the mouse colon (Fig. 3c-iv). In summary, we delineated the interaction between the mucus-adherent øPNJ-6 and MUC2 in vivo.

## Mucus-adherent phages up-regulate the expression of MUC2

To further study the interaction between øPNJ-6 Hoc and MUC2 in more detail, we examined the impact of phage or Hoc protein pre-treatment on MUC2 expression in HT-29 cells. Results showed that when HT-29 cells were exposed to øPNJ-6, we observed a transcriptional increase in MUC2 expression over the following 2 h (Fig. 3d). To investigate whether the transcriptional increase also led to the increased production of MUC2 glycoprotein, we performed Western blot (WB) analysis showing that the protein expression of MUC2 was indeed increased after 2 h of phage incubation (Fig. 3e). To further investigate whether the changes in MUC2 expression levels were caused by øPNJ-6 Hoc protein, we exposed HT-29 cells to purified Hoc protein, revealing that both MUC2 mRNA (Fig. 3f) and mucin expression (Fig. 3g) levels were both significantly increased after 2 h of co-incubation of HT-29 with øPNJ-6 Hoc protein. This provides some of the first evidence that mucus-adherent phage can facilitate the increased expression of mucin.

## Identification of glycan-binding pocket within the øPNJ-6 Hoc protein

To determine the domains of Hoc that contribute to its adherence with MUC2, we used the online tools of I-TASSER and POCASA 1.1, as well as software Pymol to predict the structure of øPNJ-6 Hoc, revealing four domains, including three Ig-like domains, domain 1(M1 – V90), domain 2 (Q95 – T185), domain 3 (K189 – A273), and a non-Ig-like domain, domain 4 (L297 – P376), which is predicted to anchor the Hoc protein to the phage capsid (Fig. 4a). We then performed prokaryotic expression to obtain the three Ig-like domains of øPNJ-6 Hoc, individually. Next, CO-IP assays were carried out to investigate the interaction between MUC2 and three Ig-like domains of Hoc. Our results indicated that structural domain 1 may play a critical role in the interaction with MUC2 (Fig. 4b), while domains 2 and 3 did not (Fig. 4c, d). We then utilized POCASA 1.1 and I-TRASSER to predict the binding sites of øPNJ-6 Hoc, identifying two protein pockets in domain 1: one at amino acids V15 – Q21 (pocket 1) and the other at E29 – G33 (pocket 2; Fig. 4a, e). We next created mutated Hoc proteins with alterations in pocket 1 (V15– E18 & T19 – Q21), to validate the predictions made by POCASA 1.1. Our results showed that Hoc protein with a mutation in pocket 1 still displayed interaction with MUC2 (Fig. 4f). This led us to speculate that pocket 2 might play a dominant role in the binding interaction. To further investigate the key amino acids of pocket 2, we expressed different mutant Hoc proteins, including mutations at E29 – G33, T30 – G32, and single point mutations of E29 or G33 (Fig. 4e). Our results showed that following insertion of mutations at E29 or G33, the interaction between Hoc and MUC2 was abolished (Fig. 4g), while other mutations did not affect the interaction between Hoc and MUC2 (Fig. 4f, g). Furthermore, we found that pocket 2 exhibited a U-shaped structure, with E29 and G33 located at the terminal of the U-shape, indicating that both E29 and G33 are the key amino acids of Hoc proteins' interaction with MUC2.

## Fucose residues determine the interaction of MUC2 and Hoc

It has been proposed that the presence of sialic acid residues at the end of the MUC2 glycan chain could protect the MUC2 from being degraded by bacterially secreted mucolytic enzymes and thus prevent the invasion of epithelial cells by intestinal bacteria[14,41,43,44]. We hypothesized that the sialic acid residues of MUC2 might be involved in the interaction between øJPN-6 Hoc and MUC2. To test this, we removed sialic acid residues for MUC2 using α2-3,6,8,9 Neuraminidase A and observed that not only did MUC2 and Hoc still interact, but that this interaction appeared to be enhanced (Fig. 4h). Along with sialic acid residues, fucose residues are also major components of the MUC2 glycan chain terminus (Supplementary Fig. 3l). Thus, we performed a similar experiment with α1-2,4,6 fucosidase O to remove the fucose residues. Our results showed that Hoc was unable to interact with MUC2 following the removal of fucose residues via pretreatment with α1-2,4,6 fucosidase O (Fig. 4i). Based on this result, we speculated that L-fucose and MUC2 would competitively bind to Hoc. To test this speculation, we conducted a competitive inhibition assay using three glycans, including sialic acid, L-fucose, which were both located at the MUC2 glycan chain terminus and Lacto-N-fucopentaose I, which was reported to interact with T4 Hoc[29]. The Hoc protein of øPNJ-6 was incubated with each of these sugars and then applied to LS174T cells. After washing with PBS, LS174T cells were collected with a lysis buffer and samples were analyzed by WB. Our results showed that binding between Hoc and MUC2 was weakened when Hoc was pre-exposed to L-fucose or lacto-N-fucopentaose I (Fig. 4j), indicating that øPNJ-6 Hoc can indeed interact with fucose residues, but not sialic acid.

To further verify the key site contributing to the interactions between Hoc and MUC2, an IFA was conducted in LS174T cells. The cells were divided into four groups: (1) positive group: Hoc + MUC2; (2) Hoc mutant E29D & G33V + MUC2; (3) Hoc + MUC2 (treated with fucosidase); and (4) negative group: MUC2. In this experiment, we fluorescently labeled our purified Hoc and its mutant protein (E29D & G33V) with FITC, followed by filtration via gravity salting-out columns to remove excess stains. The presence of both proteins was confirmed, and they were added separately to LS174T cells to assess the binding of Hoc and its mutant protein to MUC2 of LS174T. Our results demonstrated that wild-type øPNJ-6 Hoc co-localized with MUC2 (Fig. 4k-i), while the mutant Hoc protein did not (Fig. 4k-ii). The co-localization between MUC2 and Hoc also decreased when MUC2 was treated with α1-2, 4, 6 fucosidase O (Fig. 4k-iii), and there was no specific green fluorescence in the negative group (Fig. 4k-iv). Based on these results, we propose that E29 and G33 are both key sites for the interaction of Hoc with fucose residues of MUC2.

## Hoc helps phage øPNJ-6 target intestinal pathogens

Finally, we asked whether Hoc assists øPNJ-6 in adhering to gut mucins and promoting the occupation of mucosal niches in the gastrointestinal tract. To test this, we performed an inhibition assay in mice using our Hoc-blocking antibody. Mice were randomly divided into four groups: (1) øPNJ-6 + Hoc antibody + ETEC; (2) øPNJ-6 + ETEC; (3) PBS + ETEC; and (4) øPNJ-6 + PBS. Firstly, in groups 1 and 2, øPNJ-6 was pre-incubated with or without Hoc antibody followed by oral gavage

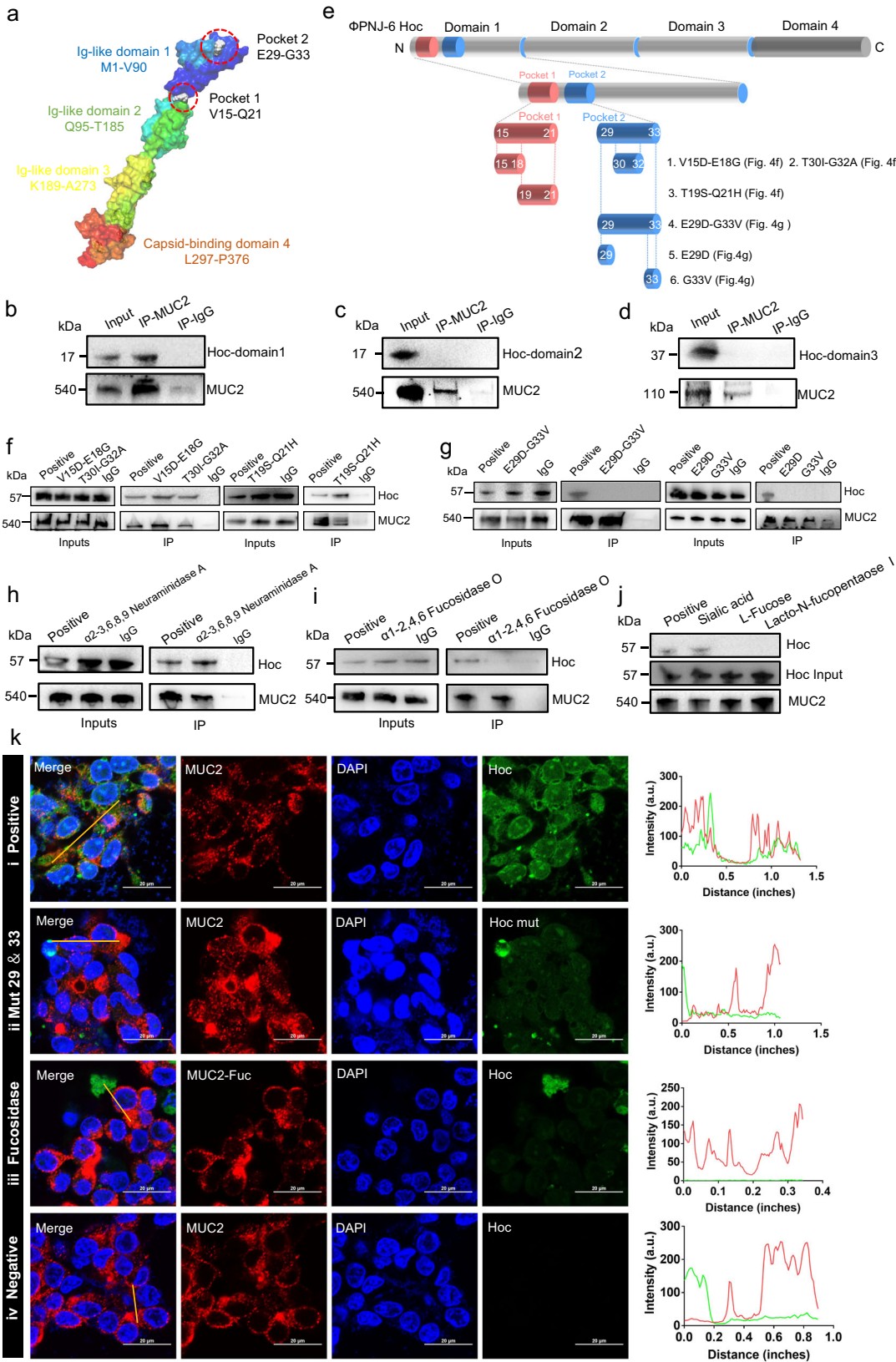

into mice to allow phage to colonize the intestinal tract for 12 h, followed by oral gavage of host bacteria ETEC. In group 3, mice were gavaged with PBS followed by ETEC. In group 4, mice were administered with øPNJ-6 followed by PBS. After 8 h, we measured the load of ETEC and the titer of øPNJ-6 in the mucus layer or the lumen of cecum and colon, respectively (Fig. 5a). In comparison to the øPNJ-6 + ETEC group, which displayed a remarkable decrease in bacterial colonization, both the antibody-coated and no-phage pretreatment groups exhibited a significant elevation in the ETEC load within the mucus layer (Fig. 5b, c) or the lumen contents (Figs. 5d, e) of the cecum and colon. Likewise, the quantity of phage presents in the mucus layer (Fig. 5f) or luminal contents (Fig. 5h) of the cecum was notably

**Fig. 4 | Interactions between glycan-binding pocket within the Hoc and fucose residues of MUC2. a** The structure of Hoc protein: contains four domains and two pockets in domain 1. CO-IP between MUC2 and domain 1 (**b**), domain 2 (**c**) or domain 3 (**d**) of Hoc protein. **e** Six mutants of Hoc were constructed in this study. (**f**) CO-IP between mutants V15D - E18G, T19S - Q21H or T30I - G32A of Hoc and MUC2. **g** CO-IP between mutants E29D - G33V, E29D or G33V of Hoc and MUC2. (**h**) CO-IP of Hoc and MUC2 treated with α2-3, 6, 8, 9 Neuraminidase A. **i** CO-IP of Hoc and MUC2 treated with α1-2, 4, 6 Fucosidase O. **j** CO-IP of Hoc pre-incubated with sialic

acid, ʟ-fucose, or lacto-*N*-fucopentaose I and MUC2. **k** Fluorescence microscope photograph of the binding of Hoc or mutant Hoc to MUC2. (**k**-i) Hoc of øPNJ-6 co-localized with MUC2 in the positive control group (×100); (**k**-ii) Mutant E29D and G33V of Hoc could not co-localize with MUC2 (×100); (**k**-iii) Hoc was unable to bind to MUC2 that was treated with α1-2, 4, 6 Fucosidase O (×100); (**k**-iv) LS174T cells in their natural state shown the absence of Hoc (×100). Red indicates MUC2, green indicates Hoc, and blue indicates cell nucleus. Scale bars, 20 µm. Source data are provided as a Source Data file.

diminished in the antibody-blocked group when compared to the other two groups. Surprisingly, the number of phage adhering to the mucus layer (Fig. 5g) or the luminal contents (Fig. 5i) in the colon demonstrated similarity between the Hoc antibody blocked and non-blocked groups, speculating that phage replication might still occur in the antibody pretreatment groups, which would have negated the initial impacts of Hoc antibody-blocking. Furthermore, it is noteworthy that the presence of both ETEC and øPNJ-6 did not yield any discernible impact on the integrity of the tight junctions comprising the gut barrier (Supplementary Fig. 3m). We also observed the presence of mucus in the lumen (Supplementary Fig. 3n), leading us to speculate that it is the primary factor contributing to the consistent change in phage numbers observed in both the lumen contents and the mucus layer following Hoc-antibody coating.

Next, to investigate whether this phage therapy model can be extended to broader intestinal pathogenic *E. coli* strains from human patients, we test the lytic ability of phage øPNJ-6 on ten *E. coli* strains (Supplementary Table 2) that were isolated from humans. The results showed that *E. coli* 029 could be lysed by øPNJ-6. Using our ETEC SH232 as the test strain, we detected the Efficiency of Plating (EOP) of 0.658 for *E. coli* 029. Strain 029, isolated from human patients' urine, was defined as a STEC strain due to the presence of the *stx2* gene (Supplementary Fig. 3o). We next asked whether the mucus-adherent phage øPNJ-6 could protect the murine intestine from STEC 029 infection. Consistent with the previous results, the intestinal bactericidal capacity of phage øPNJ-6 on STEC 029 was significantly reduced when the Hoc protein was blocked, both in the mucus layer (Fig. 5j, k) and in the luminal contents of cecum and colon (Fig. 5l, m). Correlating with this we observed a significant increase in the number phage between the antibody-blocked and non-blocked groups within the mucus layer of the cecum and colon (Fig. 5n, o), with a reduced yet non-significant difference in luminal contents (Fig. 5p, q).

Finally, to better mimic the human intestinal system, we conducted an experiment involving human isolated strain STEC 029 within the gut-on-a-chip system. During this experiment, the pump was temporarily halted after the introduction of the phage or bacteria, as previously described. Our findings revealed øPNJ-6 exhibited a higher concentration (Fig. 5r) within the mucosa, compared to Hoc antibody blocked and NAC pretreatment group, and thus led to a significant reduction in the load number of STEC 029 (Fig. 5s), accompanied by an increase in HT-29 cell viability (Fig. 5t). Taken together, these results demonstrate that the BAM model is applicable for the targeted reduction of ETEC and STEC originating from both animals and humans within the gastrointestinal mucosa. This was mediated by Hoc adherence to fucosylated mucin glycans, allowing øPNJ-6 to occupy a mucosal niche, increase the production of intestinal MUC2, and subsequently prevent pathogen invasion.

## Discussion

A previous study proposed the BAM model[30], which was further validated within a naturally-infected rainbow trout model, showing that phage binding to mucosal surfaces is important for protection against aquatic disease[32]. A subsequent study demonstrated that phages within the mucosal surface evolve to increase their affinity for fucosylated mucin glycans[29]. Here, we have investigated the BAM model in

greater depth using a phage øPNJ-6, the pathogenic bacterium ETEC from piglets, and several mucosal models, including an in vivo mouse model. We found that initial phage adherence to mucus is critical to mediate protective effects in the murine gastrointestinal tract. We identified the key binding pocket associated with phage adherence to fucose residues of MUC2. We also found that the Hoc protein and phage led to an increase in mucin expression within the gut cells, suggesting a positive feedback loop for phage adherence and mucus secretion[45]. We further extended this model to human-derived STEC, with the results being consistent with *E. coli* strains derived from animals. Moreover, the presence of Hoc in many *E. coli* phages suggests that this model may be generalizable.

In this study, we discovered that phage øPNJ-6 protects epithelial cells from *E. coli* in vivo primarily via the interaction between Hoc and MUC2. Subsequently, we further determined that Hoc primarily binds to the glycan modifications of MUC2, specifically fucose residues, which are widely distributed throughout the human intestine. Recent glycan arrays showed that Hoc protein facilitates phage binding to fucosylated glycans, with the highest affinity for Lacto-*N*-fucopentaose I, a fucosylated sugar[29]. Although the fucose residues coating MUC2 glycan chains and fucosylated polysaccharides are entirely different receptor types, they both share a common element in ʟ-fucose. We conducted a competitive assay in vitro to confirm this hypothesis and found that once ʟ-fucose on MUC2 is competitively bound, the binding of Hoc to MUC2 decreases. Fucose, a six-carbon sugar found in the mammalian gut, is a widely abundant component of glycan-decorated proteins secreted by goblet cells on the epithelial surface[46]. Besides, we found that sialic acids of MUC2 were not the binding target of Hoc protein, however, øPNJ-6 displayed enhanced adherence to MUC2 when sialic residues were enzymatically cleaved, suggesting that this treatment led to an inadvertent increase in the accessibility of fucose residue that facilitated enhanced adherence to mucus.

Previous work reported a Hoc mutation in the third domain (D246N) of T4 phage resulting in weaker binding of various fucosylated glycans compared to the wild-type Hoc of T4 phage[29]. In the present study, we found that the key binding sites of Hoc for MUC2 are E29 and G33 in the first Hoc domain of øPNJ-6. Indeed, the diversity of Hoc-glycoprotein interactions may be due to the diversity of Hoc structures. The Hoc protein is highly variable among different T4-like phage, with different phage possessing different numbers and arrangements of Hoc domains. This variability in Hoc structure likely contributes to the ability of T4-like phage to target different host species and host cells in different environments[37,47,48]. Overall, the diversity of Hoc-glycoprotein interactions highlights the versatility and adaptability of T4-like phage as a potential therapeutic strategy against bacterial pathogens in the gut.

Using a pretreatment colonization assay in both gut-on-a-chip and in vivo murine models, we demonstrate the antimicrobial efficacy of øPNJ-6 was significantly decreased when blocked by Hoc antibody, leading to higher ETEC or STEC load in the mucus layer and the luminal contents. This reduction in bacterial loads was likely the result of reduced numbers of phages adhering to mucus, indicating that Hoc plays an important role in assisting phage-mucosal targeting of host bacteria. Surprisingly, we did not observe a significant difference in the

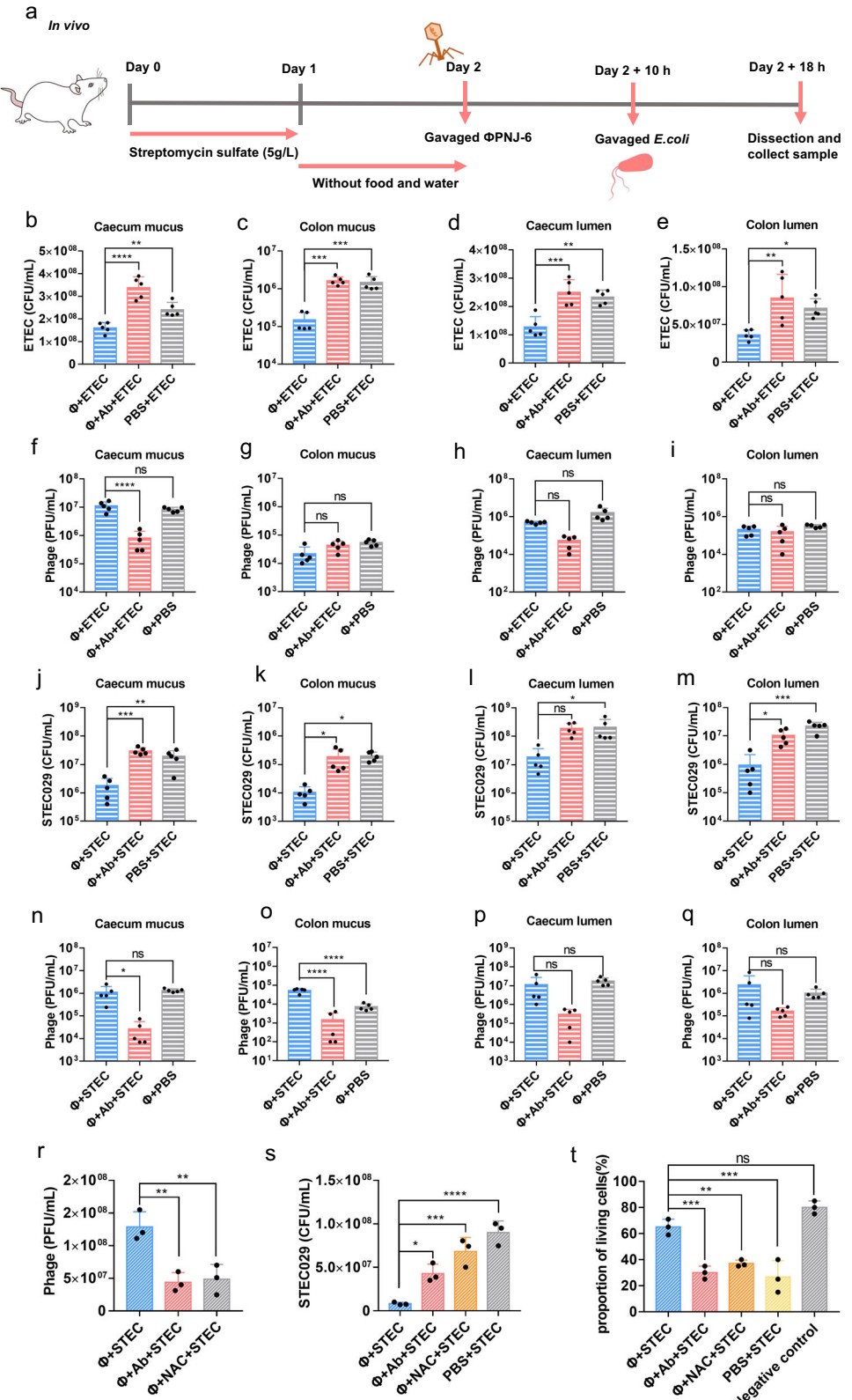

number of phage adhering to the colon of mice, with or without Hoc antibody blocking, upon co-infection with ETEC. This is likely due to the fact that the Hoc antibody blocking is only effective at blocking the Hoc domain on the initially treated phages. Once phages replicate in the gastrointestinal tract, all subsequent virions will express unblocked Hoc proteins. We speculated that phage øPNJ-6 replication occurred in vivo, since obtained results showed that phage could not be

detected in the cecum and colon at 18 h and 12 h (Fig. 2n, o) in the absence of ETEC. However, when host bacteria were present, a significant number of phages could still be detected in the colon and cecum 18 h after gavage with blocked phage (Fig. 5f–i). This may be due to phage replication in the presence of host bacteria. Consequently, we were unable to discern a noticeable difference between the two groups in terms of phage titers within the colon. Further research

**Fig. 5 | Phage øPNJ-6 specifically lysed intestinal ETEC and STEC with the help of Hoc.** In the prevention assay with ETEC in vivo: (**a**) The scheme of prevention assay in mice; The image was created by Adobe Illustrator 2023 and PowerPoint. In the groups receiving pre-treated phage or Hoc antibody-blocked phage, the abundance of ETEC SH232 was assessed in the cecal (**b**) and colonic (**c**) mucus layers, as well as the ETEC SH232 counts in the luminal contents of the cecum (**d**) and colon (**e**) in mice. The phage levels were quantified in the cecal (**f**) and colonic (**g**) mucus layers, as well as the phage counts in the luminal contents of the cecum (**h**) and colon (**i**). In the prevention assay with SETC in vivo: The presence of STEC 029 in the mucus layer of the cecum (**j**) and colon (**k**), as well as the STEC 029 counts in luminal contents of the cecum (**l**) or colon (**m**). The phage numbers were quantified in the cecal (**n**) and colonic (**o**) mucus layers, and the phage counts in the luminal contents of the cecum (**p**) and colon (**q**). In the pump-paused model with STEC in the gut-on-a-chip: phage number (**r**), STEC number (**s**), and cell survival (**t**) in the control, Hoc antibody-coated or NAC treatment groups. Data are presented as mean values ±SD. $n = 3$ biologically independent experiments. *P*-values are calculated by Multiple *t* test one per-row. *P*-values are calculated by One-way ANOVA (*$P \leq 0.05$; **$P \leq 0.01$; ***$P \leq 0.001$; ****$P \leq 0.0001$). Source data are provided as a Source Data file.

is needed to determine the full extent of phage-mediated protection of eukaryotic cells in the gut and how it contributes to the maintenance of gut homeostasis.

Our observations indicate a notable discrepancy in phage abundance between the colon and the cecum (Fig. 2h, i). The mucus layer in the colon is renewed by goblet cells every hour, and the metabolized mucus enters the rectum and is excreted with intestinal peristalsis[49], which may result in the phages attached to the mucus being easily carried away. Consequently, the number of phage adhering to the colon mucus is much lower than to the caecum mucus. Besides, we hypothesized that the pouch-like physiological structure of the cecum may be more conducive to phage retention[50], resulting in a greater difference in phage counts in the colon mucus layer than in the caecum mucus between the NAC-treated and untreated groups (Fig. 2h, i).

One limitation of this study is that HT-29 cells were cultured for a relatively short time, leading to MUC2 being detected but weak (Supplementary Fig. 1d). If the culture time was extended longer, the phage-mucus interaction observation may be more significant. Another limitation is that when we explored the important role of mucus for phage binding, the change in the number of phages in the NAC-pretreated group was weak, probably since NAC did not completely remove the mucus, but merely caused it to decrease. Using the *muc2* knockout cell line and mice may further compensate for the shortcomings of this test. Similarly, when investigating the role of Hoc, it may be more effective to use a *hoc*-deleted phage instead of Hoc antibody treatment.

In summary, we investigated the application of the BAM model in phage therapy and explored the mechanism underlining using phage øPNJ-6, ETEC from animals, and STEC from human and mouse models. We identified the key glycan-binding pocket and amino acid site of Hoc protein and found that phage or Hoc protein can up-regulate mucin expression, suggesting a positive feedback loop for phage adherence and mucus secretion. Finally, we verified that Hoc could help øPNJ-6 attach to the gastrointestinal mucus layer of mice, allowing it to occupy a mucosal niche and subsequently reduce *E. coli* colonization.

## Methods

### Ethics statement
All experiments involving mice followed the regulations of the Guide for the Care and Use of Laboratory Animals and were approved by the Institutional Animal Care and Use Committee of the Experimental Animal Center (PTA2019024).

### Bacteria strains, phage, and cell lines
SH232 was isolated from the diarrheal feces of piglets in China. Ten *E. coli* strains (Supplementary Table 2) including 029 (isolated from human urine) were gifts from Dr. Nannan Wu of CreatiPhage Biotechnology. Commensal bacteria *K.oxytoca* (bio53232) was purchased from Biobow (Beijing, China). All strains were grown in LB medium overnight at 37 °C before infection. Phage øPNJ-6 was isolated by single plaque assay from chicken fecal samples. The human colorectal epithelial cell line HT-29 was obtained from ATCC (HTB-38) and cultured in DMEM (C11995500BT, Gibco, China) medium supplemented with 10% fetal bovine serum (FBS, F2442, Sigma, Germany) and 100 μg/mL

penicillin-streptomycin (P/S, 15140-122, Gibco, China). LS174T cells were purchased from ATCC (CL-188) and seeded in MEM medium with 10% FBS and 100 μg/mL P/S.

### Genome sequencing and analysis
The genome of phage øPNJ-6 was extracted using a phage genome DNA extraction kit (ABigen, China) following the manufacturer's instructions. Subsequently, the whole genome of øPNJ-6 was sequenced and analyzed at Novogene Bioinformatics Technology Co., Ltd. (Tianjin, China). DNA libraries were constructed using the Illumina NovaSeq 6000 sequencing platform and the NEBNext Ultra DNA Library Preparation Kit (NEB, USA) for Illumina kits. First-generation sequencing was performed using an ABi 3730 (Applied Biosystems, USA) in Tsingke Biotech, Beijing, China.

### Sequence alignment analysis
Information on reference Hoc of phages (Table 1) was downloaded from GenBank. Sequence differences of Hoc between øPNJ-6 and other phages in GenBank were analyzed using the Snapgene and MEGA-X.

### Determination of phage adhesion to mucin in vitro
The ability of phage to adhere to mucin was tested both on agar plates and in the fluid medium. i) The ability of phage to bind to mucin-containing agar plates was tested according to Barr's reports[30]. LB agar plates were coated with 1% mucin and then allowed to dry. In all, 5 mL of phage ($10^3$ PFU/mL) were added into the plates with or without 1% mucin and incubated for 30 min with shaking at 37 °C. After incubation, phage suspensions were removed from the plates, which were left to dry for 30 min. Then 5 mL top agar containing 300 μL of host bacteria was added to each plate. The plates were incubated at 37 °C overnight and plaques were counted later to measure the phage held on the plates. ii) phage øPNJ-6 ($10^8$ PFU) and ETEC ($10^7$ CFU) were cultured in LB liquid with or without 1% mucin in a 1:1 volume ratio and were incubated on a 180 rpm, 37 °C shaker. Samples were collected every 2 h from 0 to 24 h to determine the phage count by double-agar plate assay.

### Measurement of mucus secretion on HT-29 cells in vitro
HT-29 cells and MDBK cells were seeded in confocal dishes until monolayer formation was achieved. The cells were fixed with Carnoy fixative (G2312, Solarbio, China) for 15 min and washed twice with PBS. Subsequently, the cells were blocked with 1% BSA (GC305010, Solarbio, China) at 37 °C for 1 h and then incubated with the primary antibody against MUC2 and the dilution ratio is 1:200 (orb547659, biorbyt, US) at 37 °C in an incubator for 1 h. After washing with PBS, the cells were further incubated with the secondary antibody and the dilution ratio is 1:500 (Goat anti-Rabbit IgG Alexa Fluor 647, ab150079, abcam, USA). Finally, the cells were stained with DAPI (KGA215, KeyGen, China) at room temperature.

### Phage lysis ability assay
Phage øPNJ-6 ($2\times10^9$ PFU, 200 μL per tube) was incubated with its polyclonal antibody to Hoc at the volume ratio of 25:1 for 1 h at 37 °C in the experimental group, and in the control group, phage øPNJ-6 was

co-incubated with mice serum. After 1 h incubation, ETEC ($2 \times 10^8$ CFU, 200 μL per tube) was added to the above tubes at 180 rpm for 1 h under a 37 °C shaker, and finally, the bacterial counts were calculated by single colony counting. The data was analysised by Graphpad prism 7.0.

## Vitro infection experiments

First, HT-29 cells were cultured in 12-well plates until a monolayer formed. The lysate liquid of phage øPNJ-6 was purified as described in previous studies[51], and then the monolayer HT-29 cells were treated with or without phage ($10^8$ PFU/mL, 1 mL per well) for 30 min at 37 °C and 5% $CO_2$. Following this, the cells were washed three times with PBS to remove any unattached phage. ETEC ($10^7$ CFU/mL, 1 mL per well) in DMEM was added to infect the cells. At different time points (3 h, 6 h, 12 h, and 24 h), the supernatant of the cells was collected to determine the number of planktonic ETEC and phage øPNJ-6. Then the cells were scraped from the plates and diluted in PBS to determine the adherent phage and bacteria, as well as the cell survival rate. If necessary, the mucus of the cells was removed by adding NAC (15 mM, HY-B0215, MCE, USA) for a 30-min incubation, followed by two washes with PBS before adding the phage. In case it was required to block phage øPNJ-6 with Hoc antibody, the phage was incubated with Hoc antibody at a volume ratio of 25:1 for 1 h at 37 °C before adding them to the cells. The viability of the cells was measured by a cell counter (Countess 3, Thermo Fisher, USA). For the competitive experiments, ETEC ($10^7$ CFU/mL) and commensal bacteria K.oxytoca ($10^4$ CFU/mL) were added in a volume ratio of 1:1 after the phage øPNJ-6 pretreatment. Finally, the indices were measured by single clone and double-agar plate assay.

## Observation of Immunofluorescence assay in vitro

To specifically target ETEC, we utilized a plasmid carrying the GFP gene to transfer into ETEC SH232 and allowed ETEC to show green fluorescence. HT-29 cells were seeded onto 15-mm glass-bottomed culture cell dishes (Nest Biotechnology, China) at a concentration of $10^6$/mL and incubated for 36 h before labeling with CellMask™ Plasma Membrane Stains (C10046, Thermo Fisher Scientific, USA) for 30 min at 37 °C. After washing with PBS, the cells were incubated with purified phage øPNJ-6 ($10^8$ PFU/mL) in DMEM media at 37 °C for 30 min. Following three washes, the cells were infected with purified ETEC ($10^7$ PFU/mL) for 3 h at 37 °C with 5% $CO_2$. The dishes were mounted with DAPI (Hoechst 33342 Stain solution, Solarbio, China) for 30 min at 37 °C and washed with PBS before being stored at 4 °C overnight for observation the next day. Fluorescence images were captured using a Nikon A1 confocal microscope (Nikon, Japan).

For the in vitro competitive assay, a plasmid carrying the Mcherry gene was transferred into K.oxtoca. A total of 1 mL bacteria containing ETEC ($10^7$ PFU/mL) and K.oxtoca ($10^4$ PFU/mL) in DMEM was added to HT-29 cells after the addition of phage øPNJ-6 ($10^8$ PFU/mL). The remaining steps were identical to the procedure described above. Fluorescence images were caputured using a Leica TCS SP8 confocal microscope with Las AFsoftware v2.7.3.9723 (Leica, Germany).

## Statistic model in vitro assay without bacteria

HT-29 cells were cultured in 12-well plates. The cells were randomly assigned to three groups: group 1 (øPNJ-6 + NAC), group 2 (øPNJ-6), and group 3 (øPNJ-6 + Hoc antibody). In group 1, NAC (15 mM) was added to the cells and incubated at 37 °C for 1 h, followed by two washes with PBS. In group 3, øPNJ-6 was blocked with its corresponding Hoc antibody for 1 h. Different pretreatments of phage were then added to the cells in the respective groups and incubated for 30 min. Finally, all cells were washed three times by PBS, and the number of phages was quantified using the double-agar plate assay.

## Established the gut-on-a-chip model

The gut-on-a-chip model used in this study was established as previously described with some modifications[31]. For all chip assays, the MOI of phage øPNJ-6 and ETEC was 10, at a concentration of $10^8$ PFU/mL and $10^7$ CFU/mL, respectively. Firstly, Extracellular matrix (ECM, E0282, Sigma, Germany) was diluted using DMEM for the ratio 1:50 and then 10 μl was added to the channel of the gut-on-a-chip for 2 h in the cell incubator. Before performing infection assays, HT-29 cells ($10^4$/mL) were cultured in the chips overnight until cells attached to the wall, and then the pump (RS-232, New Era Pump Systems, USA), which can provide dynamics, was adjusted to 45 μL/h until cells grew full in the chips, and then the rate of flow was adjusted to 120 μL/h to mimic the fluid flow and shear stresses of the human intestine[31,52]. The media (DMEM media with 10% FBS) used in the chips containing 1×MEM Non-Essential Amino Acids (MEM NEAA, 11140-050, Gibco, China), DAPT (10 μM, D5942, Sigma, Germany), and Phorbol myristate acetate (PMA, 20 nM, P1585, Sigma, Germany), which offer unnecessary amino acids and promote cells secrete mucin. To conduct the infection assay, phage ($10^8$ PFU/mL) was perfused into the chips at a rate of 120 μL/h for 50 min, followed by ETEC ($10^7$ CFU/mL) perfused into the chips at the same rate for 50 min. The cells were then cultured in the chips with flow media for 24 h. The supernatant of the cells was collected from the chip tunnel to a centrifugal tube. Trypsin was then added to the cells for 3 min at 37 °C to exfoliate the adhered cells, and all the liquid was collected in the same centrifugal tube to count the number of phage and bacteria. Cell survival rate was measured using a cell counter (Countess 3, Thermo Fisher, USA).

If it was necessary to remove the mucus of the cells in the chip, NAC (15 mM, HY-B0215, MCE, USA) was added to DMEM and perfused into the chips at a rate of 120 μL/h for 30 min before adding phage to the cells. If an antibody-coated assay was required in the chip model, the phage was incubated with their Hoc antibody for 1 h at 37 °C before being added to the cells.

## Immunofluorescence assay in gut-on-a-chip

Four experimental groups were established: group 1 (øPNJ-6), group 2 (øPNJ-6 + Hoc antibody), group 3 (øPNJ-6 + NAC), and group 4 (blank control). HT-29 cells were cultured or treated with NAC in the micro-arrays as previously described. Firstly, all samples were treated with Carnoy fixative (G2312, Solarbo, China) at room temperature for 15 min and then washed with PBS. Next, Cells were blocked with 1% BSA for 1 h at 37 °C. Primary antibodies were Soc (dilution ratio is 1:200, stored in our laboratory) and MUC2 (dilution ratio is 1:200, orb547659, biorbyt, UK) in this method assay. Finally, samples were incubated with fluorescent secondary antibodies (ab150113, abcam, UK; ab150075, abcam, UK). DAPI (Hoechst 33342 Stain solution, Solarbio, China) was used to stain the cell nucleus for 30 min at 37 °C and washed with PBS. Fluorescence images were captured using a Nikon A1 confocal microscope (Nikon, Japan). Software Image J was used to analyze the co-localization of øPNJ-6 and MUC2.

## Animal experiments

Female BALB/c mice, 4 weeks old, were used for in vivo experiments. Mice were housed in a specific pathogen-free (SPF) environment at the Laboratory Animal Center of Nanjing Agricultural University at a temperature of 22–26 °C, relative humidity of 50–60%, and alternating light and dark every 12 h.

(i) Preparation of Hoc antibody. Purified proteins were administered into mice by subcutaneous injection (100 μg for each one) for triple immunization on days 0, 14, and 28. Mouse serum was collected on day 38.

(ii) Hoc antibody blocking assay in vivo. Mice were divided into two groups equally. On day 1, mice were kept on a sterile diet with water containing streptomycin sulfate (5 g/L) for 24 h and then on a prohibited diet for another 24 h. All mice received a 5 g/L dose of

NaHCO$_3$ orally and waited for 30 min before phage treatment. In the antibody-coated group, phage øPNJ-6 (10$^9$ PFU, 200 μL/per mouse) was blocked by Hoc polyclonal antibody at a volume ratio of 10:1 for 1 h at 37 °C, while øPNJ-6 (10$^9$ PFU, 200 μL/per mouse) was incubated by mouse serum in the non-antibody-treated group at a volume ratio of 10:1 for 1 h at 37 °C, and then the phage was administered via gavage. At 12 h, 18 h, or 24 h after phage administration, the cecum, colon, and faeces of the mice were collected, homogenized, and centrifuged at 4000×$g$ for 10 min to remove pellets. The supernatant was collected and the phage number was detected via double-layer agar plates.

(iii) The experiment about the affinity of phage-to-mucin in vivo. Four-week-old female BALB/c mice were randomly assigned to three groups: group 1 (øPNJ-6) and group 2 (øPNJ-6 + NAC). Mice in group 2 were orally administered NAC at a concentration of 50 mM for 10 consecutive days. On day 9, following NAC and sodium bicarbonate (NaHCO$_3$) treatment, all groups received oral administration of øPNJ-6 (2 × 10$^8$ PFU/ per mouse), and the mice were dissected 18 h later. Luminal content and mucus layer were collected from the colon or caecum of the mice. All samples were diluted ten-fold, and the phage count was determined using a double-agar plate assay.

(iv) Prevention assay in vivo. In the mice prevention assay, 4-week-old female BALB/c mice were randomly divided into three groups: group 1 (øPNJ-6 + ETEC), group 2 (øPNJ-6 + Hoc antibody + ETEC), and group 3 (PBS + ETEC). Mice were maintained on sterile food and water containing streptomycin sulfate (5 g/L) for 24 h, followed by a prohibited diet for 24 h on day 1. To neutralize stomach acid, mice were orally administered with 5% NaHCO$_3$ (w/v) for 30 min, 200 μL each, before giving phage pretreatment on day 2. Phage øPNJ-6 (1 × 10$^9$ PFU, 200 μL per mouse) with or without Hoc antibody blocking was then orally administered to mice. ETEC SH232 carrying ampicillin resistance (1 × 10$^8$ CFU, 200 μL per mouse) was orally gavaged to mice after phage pretreatment at 10 h. Finally, mice were dissected to collect caecum and colon samples 18 h after phage pretreatment. Phages were quantified by double-layer-agar plates, and ETEC was determined by a single clone on LB plates with ampicillin. The same steps as described above were used for the STEC strains in the prophylaxis animal tests.

(v) Immunofluorescence of paraffin sections in vivo. To investigate the interaction between Hoc and MUC2 in the mice's intestinal gut, we conducted an in vivo immunofluorescence study. The mice were randomly divided into four groups: group 1, gavage with PBS; group 2, gavage with øPNJ-6; group 3, treated with NAC followed by øPNJ-6; group 4, gavage with øPNJ-6 blocked by Hoc antibody. The mice were given abundant food and water containing streptomycin sulfate (5 g/L) for 24 h, followed by a prohibited diet for 24 h on day 1. On day 2, the mice were gavaged with 200 μL of 5% NaHCO$_3$ (w/v) for 30 min. Then, in group 3, øPNJ-6 (10$^9$ PFU, 200 μL per mouse) was orally administered, while each mouse in group 1 received 200 μL of PBS. In group 2, before phage gavage, NAC (30 mM, 200 μL per mouse) was orally administered for ten consecutive days, each 24 h apart. In group 4, phage øPNJ-6 (10$^9$ PFU) was pre-incubated with its Hoc antibody at 37 °C or 1 h before gavage. Seven hours after the above treatment, the mice were dissected, and the entire colon tissues were collected and stored in Carnoy solution (G2312, Solarbio, China) at room temperature before being paraffin-sectioned. The following steps were taken for immunohistochemistry: (1) Samples were embedded in Biodewax and then washed with distilled water. (2) The slide was immersed in EDTA antigen retrieval buffer (pH 8.0) (G1206, Servicebio, China) at sub-boiling point temperature, followed by immersion in 3% H$_2$O$_2$ in the dark at room temperature. (3) The slide was blocked with serum and incubated with a primary antibody (Anti-MUC2 Rabbit pAb, 1:500 dilution ratio, GB11344-100, Servicebio, China), followed by a secondary antibody (GB21303, Servicebio, China) for MUC2. (4) The slide was incubated with CY3-TSA solution in the dark and then immersed in EDTA antigen retrieval buffer (pH 8.0) at

sub-boiling point temperature. (5) The slide was incubated with a primary antibody for Soc (Anti-Soc Mouse antibody was stored in our laboratory, 1:100 dilution ratio), followed by a secondary antibody (1:500 dilution ratio, GB27301, Servicebio, China) for Soc. (6) The slide was incubated with CY5-TSA solution in the dark. (7) DAPI (G1012, Servivebio, China) was used to stain the nucleus for 10 min in the dark, and then the slide was incubated with autofluorescence quenching reagent (G1221, Servivebio, China), and washed under running water and PBS. Finally, the slide was covered with a fade-resistant mounting solution (G1401, Servivebio, China).

(vi) Assessment of intestinal integrity by immunofluorescence assay. The assessment of intestinal integrity through immunofluorescence is similar to step v above. This entailed employing specific antibodies against E-cadherin (1:500 dilution ratio) (GB12082-100, Servicebio, China), along with their corresponding secondary antibodies (GB25301, Servicebio, China). Software Image J was used to analyze the MFI of E-cadherin in the caecum or in the colon.

## Detection of the transcription level of *muc2*

To assess the expression of MUC2, quantitative real-time PCR (qPCR) was performed by StepOnePlus real-Time PCR System (Thermo Fisher Scientific). HT-29 cells were seeded in six-well plates and treated with purified phage (10$^8$ PFU/per well) or Hoc protein (0.25 mg/per well) after 24 h. The cells were then incubated for 1, 2, or 3 h in a 5% CO$_2$, 37 °C incubator. Total RNA was extracted from the cells using an RNA extraction reagent (RC112-01, Vazyme, China) following the manufacturer's instructions, and reverse transcription was carried out using the qScript cDNA superMix (R323-01, Vazyme, China). RT-qPCR was performed using the SYBR Green Master (Q141-02, Vazyme, China), with a PCR protocol of one cycle at 95 °C for 5 min, followed by 30 cycles of 95 °C for 15 s and 55 °C for 15 s, and finally stored at 4 °C. Supplementary Table 1 shows the primers used in the experiment.

## Detection of the expression level of MUC2

HT-29 cells that have grown monolayers were treated with purified phage (10$^8$ PFU/per well) or Hoc protein (0.25 mg/per well) for 2 h in the cell incubator, while the control group was added an equal amount of PBS. Cells were lysed by NP-40 lysis solution (N8032, Solarbio, China). The expression level of MUC2 was examined by WB. Samples were boiled at 100 °C for 10 min in 1× loading buffer and then loaded onto 4–20% gels (Smart Life Science, China). Proteins were then transferred onto PVDF and blotted with antibodies against proteins of β-tublin (dilution ratio 1: 5000, AP0064, Bioworld technology, China) or GAPDH (dilution ratio 1: 5000, AP0063, Bioworld Technology, China) and MUC2 (dilution ratio 1:5000, M21002, abmart, China), following by the secondary antibodies Goat Anti-Rabbit IgG HRP with dilution ratio 1: 5000 (M21002, abmart, China). The images of WB were collected by Tanon-5200 (Tanon, China) and analysed by Image J.

## Detection of the interaction between Hoc and MUC2 by Immunoprecipitation

HT-29 or LS174T cells were cultured to form a monolayer in DMEM or MEM media (C11095500BT, Gibco, China) containing 10% FBS and 100 μg/mL P/S. Cells were lysed using lysis buffer and were scraped from plates using Coring Cell Scrapers (Sigma, Germany). To identify the interactions between the protein Hoc and MUC2, immunoprecipitation was performed. Specifically, 5 μL of MUC2 antibody (orb547659, biorbyt, UK; T56761, abmart, China) was incubated with 50 μL Protein A/G beads (B23202, bimake, USA) for 30 min at room temperature, while 5 μL of IgG antibody (B30011, abmart, China) was incubated with Protein A/G beads for 30 min at room temperature in the negative control group. Then, 2 mg of total cell lysate was added to each mixture for 8 h incubation at 4 °C shaker, followed by the addition of Hoc protein (0.4 mg). This protein complex was incubated

overnight at 4 °C. The beads were then washed 8 times with NP-40 lysis buffer (N8032, Solarbio, China) and boiled at 100 °C for 10 min in 1× loading buffer. Eluates were loaded onto 4–20% gels (Smart Life Science, China). Proteins were then transferred onto a polyvinylidene fluoride (PVDF) membrane and blotted with antibodies against proteins of Hoc (prepared by this study, dilution ratio 1:2000) and MUC2, followed by the secondary antibodies anti-mouse for Hoc (dilution ratio 1:5000, SA00001-19, Proteintech, China), and anti-rabbit for MUC2 (dilution ratio 1:5000, M21002, abmart, China). Experiments were also conducted to verify the interaction between Hoc domains or mutant Hoc with MUC2, and Hoc binding to deglycosylated MUC2 with glycosylase.

### Structural analysis and prediction of Hoc protein

First, the amino acid sequence of the Hoc protein was converted into FASTA format using SnapGene. Hoc sequence in FASTA format was then submitted to the I-TASSER website (https://zhanggroup.org/), which can predict protein structure. The prediction model with the highest C-score and TM-score greater than 0.5 was selected to ensure the correct topology. The online tool POCASA was utilized to predict the protein pockets present in the Hoc protein structure. Pymol was used to analysis the structure of Hoc and protein pockets.

### Prokaryotic expression of Hoc, Hoc mutants, and domains of Hoc

All proteins, including Hoc, Hoc domains, and Hoc mutants, were obtained via prokaryotic expression using plasmids pET-28a-*Kana*+ or pET-32a-*Amp*+. Primers were designed based on the sequence range of domain 1 of Hoc from M1 to T98, domain 2 of Hoc from Q91 to P193, and domain 3 from N187 to L297 using an online tool (https://crm.vazyme.com/cetool/simple.html). All primer sequences are listed in Supplementary Table 1. The constructed plasmids were expanded using DH5a and the proteins were expressed using BL21 by induction with Isopropyl-beta-D-thiogalactopyranoside (IPTG) (I8070, Solarbio, China) overnight at 16 °C and 160 rpm. The proteins were then purified using GE Cytiva columns (17524701, Cytiva, USA) and AKTA Pure (GE Cytiva, Sweden).

### Fixed-point mutation of amino acids

First, primers were designed using the online tool: https://crm.vazyme.com/cetool/singlepoint.html. The target genes were then amplified using PCR (nexus gradient, Eppendorf, Germany) with the following protocol: 25 μL of high-fidelity enzymes (P520-02, Vzyme, China), 2.5 μL of forward primer, 2.5 μL of reverse primer, template plasmid (2.5 μL, 3 ng/μL), and 17.5 μL of ddH₂O. The PCR procedure was as follows: 95 °C for 3 min, (95 °C for 15 s, 60 °C for 15 s (depending on primer TM value), 72 °C for 30 s) × 30 cycles, 72 °C for 5 min, and 4 °C for storage. Next, the plasmid was cleaved with Dpn I (1235 S, Takara, Japan) at 37 °C for 1 h. The product was then transferred to DH5a by chemical transformation. The subsequent steps were consistent with the steps for prokaryotic expression of the protein. All primers are listed in Supplementary Table 1.

### Remove the terminal glycan residues of MUC2

MUC2 was obtained from LS174T cell lysates using a protein A/G bead-based immunoprecipitation approach. Specifically, protein A/G beads (B23202, bimake, USA) were incubated with MUC2 antibody for 30 min at room temperature, followed by the addition of LS174T lysate and overnight incubation at 4 °C with rotation. After the capture of MUC2 by the protein A/G beads, the terminal sialic acid or fucose residues were removed by treatment with α 1-2,4,6 fucosidase O (P0749S, NEB, USA) or α 2-3,6,8,9 Neuraminidase A (P0722S, NEB, USA). This step was performed by incubating the enzyme with the protein complex at 37 °C for 1 h with rotation.

### Detection of the interaction between Hoc and MUC2 by Immunofluorescence Assay

To investigate the specific site of interaction between Hoc and MUC2, we conducted the following experiment. First, purified Hoc protein was labeled with FITC (F7250, Sigma, Germany) and excess dye was removed using a prepacked desalting column (SEC003C2, Smart-Lifescience, China). LS174T cells were cultured in 15-mm glass-bottomed culture cell dishes (Nest Biotechnology, China) for IFA. FITC-labeled Hoc protein was added to the dishes and incubated for 1 h at 37 °C. After washing twice with PBS, the cells were fixed with 4% paraformaldehyde at room temperature for 10 min, followed by washing twice with PBS for 5 min each. Cells were blocked with 1% BSA and then incubated with a MUC2 antibody (orb547659, Biorbyt, UK) and a secondary antibody (ab150079, Abcam, USA). Finally, the nuclei of the cells were stained with DAPI (KGA215-10, KeyGen, China). To remove sugar residues from MUC2, LS174T cells were treated with α1-2,4,6 fucosidase O (P0749S, NEB, USA) for 1 h at 37 °C before adding Hoc protein. The same method was used to verify the interaction between E29D and G33V of Hoc protein with MUC2.

### Hoc interacts with three different monosaccharides in vitro

To investigate the binding affinity of Hoc with different types of glycans, we conducted the following experiment. LS174T cells were cultured in a 12-well plate for 24 h and treated with sodium butyrate (1 mM, S1999, Selleck, USA) to stimulate the secretion of MUC2 for 36 h. Purified Hoc protein was then separately incubated with sialic acid (10 mM, A0812, Merck, Germany), L-fucose (10 mM, L809666, Marcklin, China), or lacto-N-fucopentaose I (10 mM, GY1147, Glycarbo) at 37 °C for 1 h. These mixtures were added to LS174T cells, which were subsequently incubated in a 5% CO₂, 37 °C incubator for another 1 h. After washing the cells with PBS twice, they were collected using NP-40 lysis buffer, and Hoc antibody was used to detect the samples by WB.

### Pathotype and virulence genes detection

Multiplex PCR was performed to identify pathotypes of *E.coli* strains as previously described[53,54]. If the gene of the *stx* gene is positive, it is STEC[53]; If the gene of the *STa, STb*, or *LT* is positive, it is ETEC[55–58]; if the gene of the *aggR* or *astA* is positive, it is enteroaggregative *E. coli* (EAEC); if the gene of the *bfpB* is positive, it is enteropathogenic *E. coli* (EPEC); if the gene of the *invE*, it is enteroinvasive *E. coli* (EIEC)[54]; Virulence genes including *K99*, *eae*, *hly*, and *ipah* were detected using PCR[59]. All primers were listed in Supplementary Table 1.

### Alcian blue staining of Mucin

To perform Alcian Blue staining, an appropriate number of HT-29 cells were cultured in small confocal dishes for 48 h. Purified Hoc protein (0.5 mg per dish) was added to the HT-29 cells and incubated at 37 °C for 6 h. After the incubation period, all cells were treated with Carnoy fixative (G2312, Solarbo, China) at room temperature for 15 min. The mucin layer of the cells was then stained with Alcian Blue (A8140, Solarbo, China) according to the instructions provided in the manufacturer's manual. Finally, all samples were observed using a confocal microscope (A1, Nikon, Japan).

### Pause flow rate in gut-on-a-chip

HT-29 cells were cultured and treated in a gut-on-a-chip system following the aforementioned protocol. To facilitate phage attachment to the cells, the flow rate was temporarily halted by stopping the pump for 30 min after the addition of phage (1×10⁹ CFU, 100 μL). Subsequently, a fresh medium without phage or bacteria was introduced into the cells using the pump for 30 min. The medium, containing either ETEC SH232 (1×10⁸ CFU, 100 μL) or STEC029 (1×10⁴ CFU, 100 μL), was then flowed into the cells for 50 min. Following this, the pump was paused for another 30 min to allow the bacteria to infect the cells. The cells were subsequently cultured with fresh medium for 24 h.

Finally, the cells were collected, and their viability, as well as the bacterial and phage load, were assessed. In the NAC (15 mM, HY-B0215, MCE, USA) pretreatment group, NAC was added according to the aforementioned procedure. After NAC pretreatment, the pump was paused for 30 min. In groups where only phage was present without bacteria, the step involving the addition of bacteria was omitted, while the remaining steps remained the same as described above.

## Statistical analysis and reproducibility

Statistical analysis was performed using GraphPad Prism Software (GraphPad Software, La Jolla, CA, USA). Error bars represent 5–95% of the confidence interval, and the middle line represents the mean ± standard deviation (SD) of the bar and line graphs. The results were evaluated by unparid $t$ test, One-way or Two-way analysis of variance (ANOVA). $P \leq 0.05$ was considered significant. Significance is indicated in the figures by asterisks ($*P \leq 0.05$; $**P \leq 0.01$; $***P \leq 0.001$; $****P \leq 0.0001$). Experiments that related to micrographs were repeated at least three times and similar results were obtained.

## Reporting summary

Further information on research design is available in the Nature Portfolio Reporting Summary linked to this article.

## Data availability

Phage PNJ-6 genome sequence were uploaded to NCBI GenBank under accession code OQ076693.1. Sequencing results of Hoc, Hoc mutants, Soc, and virulence factor of ETEC SH232 or STEC 029 are published and can be found in SRA data: PRJNA1014201. Source data are provided with this paper.

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

## Acknowledgements

This study was supported by the National Key R&D Program of China (2023YFD1800300) and National Natural Science Foundation of China (32172858). We thank Dr. Nannan Wu of CreatiPhage Biotechnology for giving us 10 *E.coli* strains including 029 from humans, Professor Huanming Xia of Nanjing University of Science and Technology for providing us with a platform for chip fabrication, and Zhiyu Shi from the instrument platform of Institute of Immunology, College of Veterinary Medicine, Nanjing Agricultural University for assistance in using laser confocal microscope (Nikon A1).

## Author contributions

Fang Tang and Jeremy J. Barr designed the experiments. Jianjun Dai provided valuable suggestions for the manuscript. Jiaoling Wu wrote the manuscript and performed most of the experiments described in the manuscript. Kailai Fu, Chenglin Hou, Yuxin Wang, Chengyuan Ji, Feng Xue, and Jianluan Ren offered help during the experiments. All authors have read and approved the final version of the manuscript.

## Competing interests

The authors declare no competing interests.
