## [Peer Review File · Nature Communications]

REVIEWER COMMENTS

Reviewer #1 (Remarks to the Author):

These researchers address an interesting question based on a series of their own PNAS papers from 2013, 2015 and 2022. In the 2013 paper they developed a thoughtful model where the mucus layer overlaying the mucosa in animals across a wide taxonomical range binds bacteriophages that protect the mucosa from bacterial adherence or invasion via phage-mediated lysis of bacteria. In their 2015 paper they associated the T4 phage capsid protein Hoc with the mucin glycan binding activity of the phage using a microfluidic device (chip) as test system. Hoc+ T4 in contrast to mucus-non-binding Hoc- T4 phage reduced the colonization of the chip with target bacteria by 3-logs paradoxically without differing in accumulation and persistence of the phage on the chip. The authors explained this paradox by the observation of a distinct “subdiffusive” motion in the mucus for the phage and its mutant, respectively. In their 2022 paper the authors did evolutionary studies on these chips investigating the tri-partite interaction between phage-bacterium-mucus producing cell. They identified a mutation in the third Ig domain of the Hoc protein that mediated the interaction with fucosylated mucin glycans.

The present submission is a continuation of this research line. It remains however unclear why the authors have changed the better defined T4 phage system with its available phage mutants and previous research results for a new T4-like phage experimental system. It remains therefore unclear what previously established results from T4 apply to the new phage system. Some terminology is not consequently used: for example enteropathogenic and enterotoxigenic *E. coli*, well distinguished in the literature, are interchangeably used. It is also distracting that mucosal cells and mucus layer are not everywhere clearly distinguished. Histologists distinguish two layers of mucus, one gut luminal-oriented less densely meshed mucus layer with nests or microcolonies of bacteria and below a denser less populated mucus layer; both mucus layers are highly dynamic acting as conveyer belts with loss of material into the gut lumen and translation towards the anus. On the mucosa surface hardly any bacteria are seen and the consensus is that only some gut pathogens have learned to reach the mucosal cell layer in vivo. In addition the epithelial layer is rapidly renewed with constant desquamation of cells (perhaps as a mechanism to remove bacteria that succeeded to reach the mucosal epithelial layer).

With respect to the experiments:

L.97/98: The enrichment of phi PNJ-6 in the mucus layer is not convincingly demonstrated since it needs a better demonstration than binding of phage to mucin coated vs. non-coated plates. One would wish here in vivo transit data in mice where gut segments are dissected, luminal content washed out, then the segments laterally opened and the mucus and mucosal layer scraped off. Phage titers should then be determined in the different sections and fractions and with these data one could then more convincingly assess whether the phage has a specific affinity for the mucus layer (higher abundance in the mucus than in the lumen) or whether this is just a sticking minor fraction of the orally applied phages (reminding the Roman quote that *aliquid semper haeret*). The lack of a Hoc- mutant is here a clearly missing point.

L. 104 mucin production by HT29 cells? As these are short experiments (days), was the time period sufficient to allow differentiation of HT29 cells to express mucins? A control experiment with MUC staining as depicted in Fig. 3c for the mouse colon would be helpful.

L. 107: could this be an effect of phage lysis in the supernatant?

L. 108: meaning not clear.

L. 103-115: here a control experiment with a HOC minus mutant (as previously done for T4) would be a good control.

Fig. 1a,b: growth of ETEC with and without phage (fitting the story line of the manuscript?). After 24 h no phage protection on cell viability was seen, all cell cultures were dead. Does LPS contaminate the phage preparation leading to toxic effects on host cells when not washed away by fluid flow?

Fig. 1 d,e: only 1-log ETEC reduction (much less than the 3-log reduction reported previously) but a massive effect on cell viability – explanation?

The difference between 1a,b and 1de is disturbing pointing to effects of the experimental system.

Fig. 1k: why should K oxy be higher in presence of phage?

L. 126: the conclusion is not clear since no interaction between phage and intestinal host cells was demonstrated.

Fig. 2 a-c, panel a shows a scheme, experimental data would here be better. Taken at face value, NAC pre-treatment had no effect on phage and ETEC binding which would suggest that mucus presence is not an important determinant which contradicts the main story line of the submission. Likewise, pretreatment of phage with Hoc antibody had no effect on phage and ETEC binding which contradicts the main story line developed for Hoc's role later in the manuscript. Explanations? Fig. 2 g,h shows a NAC effect, albeit a very small one (a 2- to 3-fold decrease in phage attachment) when conducted in vivo in mice (physiologically relevant?).

Fig 2n,o: could this represent clearance of antibody-coated phage in vivo?

Fig. 2 j,l contradictory outcome in the two experimental systems, interpretation?

L. 199-200 is not a good explanation.

L. 225, wasn't it said on L. 307-8 that Hoc ab did not neutralize the phage infectivity in vivo?

Fig. 5 b,c and 5 f,g demonstrate only a less than 1-log reduction of the orally applied ETEC. Can they exclude that this small reduction is mediated by the residual luminal phage in the mouse gut? Fig. 5 d,e and 5 h,i lack the phage alone control which would allow such an assessment (distinction of luminal vs. mucus-adherent phage would here be important), particularly since a high dose of 10×10^9 phage was given to a mouse, 10-times more than the challenging ETEC dose.

L. 392-395: extension of the results on phage therapy is an over-interpretation of the modest less than 1-log ETEC reduction in mice (and the in vitro systems) and lack of attenuation for diarrheal disease in mice (not a model for ETEC diarrhea compared to infant rabbits or pigs).

Reviewer #2 (Remarks to the Author):

The authors have conducted a comprehensive study using an in vitro cell culture model, an intestinal microfluidic model, and an animal model to uncover the interaction of an E. coli phage that binds mucin through its Hoc protein. This interaction leads to increased mucus production, enhanced adhesion in the gut, and ultimately impedes the colonization of E. coli, which is an intriguing piece of research. While the manuscript presents valuable experimental work and conclusions, there are a few areas that could be refined:

In lines 123-124, the authors mention that the phage-pretreated group in the gut-on-a-chip model can

survive normally for up to 24 hours, whereas in the in vitro model, almost all of them die after phage treatment. This discrepancy raises questions. Is the difference in medium flow in the gut-chip influencing the outcome? Additionally, at the end of this paragraph (lines 125-127), it's suggested that the primary reason for the reduced bacterial number is the phage's pre-lysis of the bacteria. However, the text initially discusses the interaction between host cells and bacteriophages. It would be helpful to clarify the basis for this sequence.

Figure 1 requires image quality improvement. Fig. 1c lacks clarity, making it hard to discern the scale. Fig. 1i has non-uniform image sizes, and the indicator color for *K. oxytoca* is difficult to distinguish. Consider splitting the channels or using more distinct colors.

In lines 165-171, the authors mention contradictory conclusions between the intestinal chip and the in vitro model. The purpose of this statement should be clarified. Does it imply that the intestinal microfluidic system better mimics the real situation? Although the advantages of the gut-chip system are outlined in line 200, it would be beneficial to provide a detailed explanation earlier. Additionally, the results in Figure 1 obtained by the two methods are consistent, but in Figure 2, they are contradictory. The significance of using two models may need further clarification.

The article contains some errors in the text, and it's recommended to conduct a thorough proofreading, including reviewing the figure legends.

The text would benefit from further refinement and elaboration for clarity and coherence.

These improvements will enhance the quality and readability of the manuscript.

Reviewer #3 (Remarks to the Author):

The manuscript by Wu et al. delves into the utilization of a novel mucus-adherent phage, ϕ PNJ-6, in the context of phage therapy for gastrointestinal infections. Through a combination of conventional in vitro models, a microfluidic gut model, and in vivo murine experiments, the research underscores the remarkable capacity of ϕ PNJ-6 to enhance gastrointestinal persistence and exert antimicrobial effects. It accomplishes this by binding to fucose residues in the MUC2 glycoprotein, which, in turn, stimulates increased production of intestinal mucus. Ultimately, this investigation expands the application of the BAM model to phage therapy, confirming its in vivo effectiveness and offering valuable insights for future approaches to intestinal phage. Given the current advances of the organ-on-chip systems and the ever-growing interest in the human intestinal microphysiological systems, as well as the potential of phages to treat diseases associated with gut (microbiota), the manuscript is very timely and would likely attract broad interest. However, I have some concerns regarding the model and reported results as detailed below:

- ETEC colonization: The authors introduced the ETEC in the dynamic phase along with the flow of the culture medium through the pump, which might reduce the adherence of the pathogens. Instead, the authors should let the ETEC sit in the static phase for a couple of hours and then introduce the flow. This technique has been reported previously for the inoculation of pathogens on a chip (DOI: <https://doi.org/10.1186/s40168-019-0650-5>). The same argument is valid for adding *E. coli* and phages on the chip. Part of the reason for the better results on the chip most likely comes from the fact that the bacteria do not have time to interact with the host due to the approach used by the authors. In contrast, in the in vitro setting, they have ample time to adhere and colonize. Therefore, this may not be a fair

comparison between the groups.

Challenges with the HT-29 model: Mucus secretion by HT-29 cells is not substantial, and the immunofluorescence (IF) and Western blot images for MUC2 are not satisfactory. The IF images have a lot of background noise (Fig 3g). Additionally, it is very unlikely that these cells secrete a significant amount of mucus during the relatively short gut chip culture period used in this study. The authors should consider staining the mucus with other reagents such as Wheat Germ Agglutinin or Alcian Blue. I recommend using primary or iPSC-derived intestinal organoids as a better model. These models not only secrete more physiologically-relevant mucus but also provide a better mimicry of the human response.

Measuring barrier function: Authors should measure the barrier function of the intestinal epithelium in the presence of ETEC and ϕ PNJ-6. This would provide a better understanding of the integrity of the epithelium and host-pathogen interactions.

Gut-on-a-chip images are missing: While the authors have found different results on the Gut Chip platform, all the IF images in the manuscript focus on in vitro static culture. It is essential to include IF images for the Gut Chip to provide a complete picture.

Species-specific pathogenicity of ETEC/E. Coli: It's worth noting that there are species-specific differences in the pathogenicity of ETEC/E. Coli. The real strength of the organ-on-chip system is its potential to offer a more realistic platform for human studies compared to many mouse models. The conclusions drawn about the effect of phage therapy on human E. coli strains and the role of Hoc in phage attachment have primarily been made using animal models. The authors have an opportunity to validate these findings using their Gut Chip system. Many of the observations from mice models may not necessarily translate to human patients, and the human Gut Chip platform could be instrumental in addressing this gap.

Below we provide detailed responses (in blue) to each point made by the referees.

Reviewer comments:

Reviewer #1 (Comments for the Author):

These researchers address an interesting question based on a series of their own PNAS papers from 2013, 2015 and 2022. In the 2013 paper they developed a thoughtful model where the mucus layer overlaying the mucosa in animals across a wide taxonomical range binds bacteriophages that protect the mucosa from bacterial adherence or invasion via phage-mediated lysis of bacteria. In their 2015 paper they associated the T4 phage capsid protein Hoc with the mucin glycan binding activity of the phage using a microfluidic device as test system. Hoc+ T4 in contrast to mucus-non-binding Hoc- T4 phage reduced the colonization of the chip with target bacteria by 3-logs paradoxically without differing in accumulation and persistence of the phage on the chip. The authors explained this paradox by the observation of a distinct “subdiffusive” motion in the mucus for the phage and its mutant, respectively. In their 2022 paper the authors did evolutionary studies on these chips investigating the tri-partite interaction between phage-bacterium-mucus producing cell. They identified a mutation in the third Ig domain of the Hoc protein that mediated the interaction with fucosylated mucin glycans.

The present submission is a continuation of this research line. It remains however unclear why the authors have changed the better defined T4 phage system with its available phage mutants and previous research results for a new T4-like phage experimental system. It remains therefore unclear what previously established results from T4 apply to the new phage system.

Reply: We sincerely appreciate your valuable comments. Regarding our use of the newly isolated phage ϕ PNJ-6 in place of the previously characterized T4 phage, this was chosen for several reasons. First, while the T4 phage system has been incredibly useful in establishing the BAM model and its mechanisms, there are inherent limitations with continuing this research purely focused on this phage. Second, the isolation of a novel phage with mucus-adherent properties, further extends the application and utility of the BAM model. And third, our study intended to test the clinical application of a mucus-adherent phage within the gastrointestinal tract. Here there are obvious limitations with the use of T4-phage, as it is largely unable to infect these clinically relevant ETEC and STEC strains.

Our study has provided significant experimental evidence that ϕ PNJ-6 is a mucus-adherent phage, that this phage is capable of reducing bacterial load and protecting the underlying epithelium from bacterial challenge, and has demonstrated the mechanism of mucosal adherence. Taken together, these results further extend the clinical utility of the BAM model, well beyond what we would have been able to achieve using T4 phage alone.

Some terminology is not consequently used: for example enteropathogenic and enterotoxigenic E. coli, well distinguished in the literature, are interchangeably used. It is also distracting that mucosal cells and mucus layer are not everywhere clearly distinguished. Histologists distinguish two layers of mucus, one gut luminal-oriented less densely meshed mucus layer with nests or microcolonies of bacteria and below a denser less populated mucus layer; both mucus layers are highly dynamic acting as conveyor belts with loss of material into the gut lumen and translation towards the anus.

On the mucosa surface hardly any bacteria are seen and the consensus is that only some gut pathogens have learned to reach the mucosal cell layer *in vivo*. In addition the epithelial layer is rapidly renewed with constant desquamation of cells (perhaps as a mechanism to remove bacteria that succeeded to reach the mucosal epithelial layer).

Reply: We have gone through the manuscript carefully and distinguished enteropathogenic and enterotoxigenic *E. coli*, as well as mucosal cells and mucus layer.

With respect to the experiments:

1. L.97/98: The enrichment of phi PNJ-6 in the mucus layer is not convincingly demonstrated since it needs a better demonstration than binding of phage to mucin coated vs. non-coated plates. One would wish here *in vivo* transit data in mice where gut segments are dissected, luminal content washed out, then the segments laterally opened and the mucus and mucosal layer scraped off. Phage titers should then be determined in the different sections and fractions and with these data one could then more convincingly assess whether the phage has a specific affinity for the mucus layer (higher abundance in the mucus than in the lumen) or whether this is just a sticking minor fraction of the orally applied phages (reminding the Roman quote that *aliquid semper haeret*). The lack of a Hoc-mutant is here a clearly missing point.

Reply: We sincerely appreciate your valuable comments and wholeheartedly agree with your insightful suggestions. To address this, we have conducted additional experiments as follows:

The female Balb/c mice were administered with ϕ PNJ-6 (2×10^9 PFU/mL, 200 μ L per mouse) via gavage. After a 7-h phage pre-treatment period, the mice were dissected, and luminal contents were collected. Subsequently, any remaining remnants within randomly selected 2-cm lengths of the entire caecum or colon were thoroughly washed out. The corresponding mucus layers were carefully scraped off and collected. Phage titers were determined using the double-agar plates assay. The results showed that the phage counts were found to be approximately equal (around 1×10^5 PFU/mL) in both the lumen and the mucus layer of the caecum and colon (Fig.1 in this letter, not shown in the revised manuscript). Previous studies have suggested that luminal contents also contain mucin, similar to the shedding of the upper layer of mucus that coats the surface of contents in the luminal tract (doi: 10.1128/mBio.03474-20). Therefore, we speculate that this phenomenon may be attributed to the presence of mucin in the luminal contents.

Fig 1.

To further investigate the affinity of phage to the mucin *in vivo*, we administered phage ϕ PNJ-6 orally to mice with and without mucus. To remove the mucus, mice were orally administered with NAC (50 mM) for a continuous period of 10 days. In this experiment, mice were randomly divided to two groups: (1) ϕ PNJ-6, (2) NAC+ ϕ PNJ-6. The number of phage in the mucus layer and the

lumen were measured separately. The results revealed significant decrease in phage titers in the NAC pretreatment group compared to ϕ PNJ-6 treated group, both within the mucus layer of the caecum and colon (Fig. 2h ~ 2i in the lines 226-231 of manuscript with revised mark), as well as in the luminal contents (Fig. 3a, b in this letter, not shown in the manuscript). This results clearly suggested that phage exhibit an affinity for mucus.

Fig. 2h ~ 2i in the revised manuscript

Fig. 3a, b in this letter (not shown in the revised manuscript)

We acknowledged that the absence of the Hoc-mutant group was a limitation in our study, however, we cannot construct a Hoc deleted mutant in a short time. Thus, we blocked the hoc protein by antibody as an alternative. The results of Hoc blocking group also clearly suggested the interaction between hoc and mucin.

2. L. 104 mucin production by HT29 cells? As these are short experiments (days), was the time period sufficient to allow differentiation of HT29 cells to express mucins? A control experiment with MUC staining as depicted in Fig. 3c for the mouse colon would be helpful.

Reply: In our study, prior to conducting the *in vitro* experiments, HT-29 cells were already cultured in 12-well plates for a duration of 60 h to establish an appropriate cellular environment. To address this concern, after culturing the cells for 60 h, we conducted an immunofluorescence assay to assess mucin production specifically by HT-29 cells, while Madin-Darby Bovine Kidney (MDBK) cells were utilized as a negative control. Through microscopic examination, we observed the presence of MUC2 in HT-29 cells, whereas MUC2 was barely detected in MDBK cells (Supplementary Fig. 1d in the lines 117-119 of manuscript with revised mark).

Supplementary Fig. 1d in the revised manuscript

3. L. 107: could this be an effect of phage lysis in the supernatant?

Reply: In our experimental protocol, ϕ PNJ-6 was incubated with HT-29 cells for a duration of 30 min. Then to remove any unattached phages, the supernatant was eliminated by two washes with PBS before inoculation of ETEC. Thus, the effect of phage lysis in the supernatant should be negligible.

4. L. 108: meaning not clear.

Reply: We have made revisions to the sentence as indicated in lines 129-132 of the revised manuscript.

5. L. 103-115: here a control experiment with a HOC minus mutant (as previously done for T4) would be a good control.

Reply: We acknowledge that utilizing the Hoc-mutant ϕ PNJ-6 in the experiment would have been advantageous, but we cannot construct a Hoc minus mutant in such a short time. However, we would like to highlight that we observed a significant effect when Hoc was blocked using its corresponding antibody. We believe that this finding compensates for any potential limitations in the experimental design.

6. Fig. 1a,b: growth of ETEC with and without phage (fitting the story line of the manuscript?). After 24 h no phage protection on cell viability was seen, all cell cultures were dead. Does LPS contaminate the phage preparation leading to toxic effects on host cells when not washed away by fluid flow?

Fig. 1a, 1b: To investigate whether LPS contaminate the phage preparation led to toxic effects on host cells, we conducted an additional experiment to provide further clarification. In this experiment, the phage lysate was incubated with HT-29 cells for 3 h, 6 h, 12 h, and 24 h, and the cell viability was assessed. The results showed that phage lysate did not harm the cells even incubated for 24 h (Supplementary Fig. 1e in lines 120-121 of the revised manuscript with revised mark). Therefore, we propose that the observed decrease in cell viability in the *in vitro* experiment is more likely attributed to the exhausted of nutrients in the medium, rather than contamination in the phage preparation.

Supplementary Fig. 1e in the revised manuscript

Fig. 1 d,e: only 1-log ETEC reduction (much less than the 3-log reduction reported previously) but a massive effect on cell viability – explanation?

The difference between 1a,b and 1de is disturbing pointing to effects of the experimental system.

Reply: This is an important point to emphasize regarding the differences between our experimental models. The data presented in Fig1a,1b was performed on static tissue culture cells. In these experiments, the microbial growth within the supernatant (ie., tissue culture media) rapidly depletes the nutrients available to the cells, resulting in cell death and higher bacterial growth at later time points (e.g., 12 & 24hrs). The gut-on-a-chip model largely overcomes these issues with the use of a pump system that continually washes away microbes in the supernatant and provides replenished nutrients to the system and cell layer. Regarding the 1-log reduction in ETEC coupled with a large effect on cell viability, we would explain this due to the increased pressure to attach and replicate within the mucosal surface, leading to higher bacterial-cell interactions, which potentially result in higher cell death than the *in vitro* static cultures. As such, we believe the gut-on-a-chip model is a more accurate representation of the phage-bacterial dynamics seen *in vivo*.

Specifically, regarding the differences between our experiment systems, these should be considered together in light of the above points. We have expanded our results to specifically address these points (see lines 277-284 in the manuscript with revised mark).

Fig. 1k: why should K oxy be higher in presence of phage?

Fig. 1k: The dynamic predation by ϕ PNJ-6 to reduce ETEC, coupled with the competition for nutrient access, likely results in increased growth of *K. oxytoca* while in the presence of anti-ETEC phage.

7. L. 126: the conclusion is not clear since no interaction between phage and intestinal host cells was demonstrated.

Reply: We agree with your suggestion and have revised the sentence in lines 146-148 in the manuscript with revised mark.

Fig. 2 a-c, panel a shows a scheme, experimental data would here be better. Taken at face value, NAC pre-treatment had no effect on phage and ETEC binding which would suggest that mucus presence is not an important determinant which contradicts the main story line of the submission. Likewise, pretreatment of phage with Hoc antibody had no effect on phage and ETEC binding which contradicts the main story line developed for Hoc's role later in the manuscript. Explanations? Fig. 2 g,h shows a NAC effect, albeit a very small one (a 2- to 3-fold decrease in phage attachment)

when conducted *in vivo* in mice (physiologically relevant?).

Fig 2n,o: could this represent clearance of antibody-coated phage *in vivo*?

Fig. 2 j,l contradictory outcome in the two experimental systems, interpretation?

Reply:

Fig 2a-c: Following your suggestions, we changed original Fig. 2a and Fig. 2i (figure of scheme) to Supplementary Fig. 2h in the revised manuscript.

Regarding the contradictory results observed across the two models. We have carefully considered the possibility that the presence of ETEC may be responsible for the divergent outcomes in the two experiments. To address this, we conducted an additional experiment to elucidate this phenomenon in the absence of bacteria both in *in vitro* and gut-on-a-chip models. In this new experiment, HT-29 cells were either pre-treated with NAC for 30 min, followed by incubating with ϕ PNJ-6 or incubated with Hoc antibody blocked phage. *In vitro*, the results demonstrated phage titer decreased significantly after NAC pre-treatment (Supplementary Fig. 2k in lines 202-208 of the manuscript with revised mark) or antibody coated phage pre-treatment (Supplementary Fig. 3f in lines 254-259 of the manuscript with revised mark) compared to ϕ PNJ-6 pre-treated group. The results in the gut-on-a-chip are consistent with that of *in vitro* model (Supplementary Fig. 2l in lines 210-213, Supplementary Fig. 3g in lines 256-260 of the manuscript with revised mark).

Supplementary Fig. 2k, 2l in the revised manuscript

Supplementary Fig. 3f, 3g in the revised manuscript

Fig 2n, o: There is potential for these antibody-treated phage to be cleared by the body and the immune system, but this would require secretion into the intestinal lumen or uptake of antibody-treated phages by the gut epithelium. A more likely explanation, which fits with our data presented to date, is that antibody-blocked ϕ PNJ-6 have a lower residence time in the gut due to blocked

interaction with gut mucins.

Fig2j,l: Please see our responses above. These differences are due to the two experimental models being used (static tissue culture versus gut-on-a-chip). We have provided further in-text summary to help explain this further (see lines 277-284 in the manuscript with revised mark).

8. L. 199-200 is not a good explanation.

Reply: We have removed the sentence and added new experiments to explain it in the revised manuscript. Please see our reply above (lines 277-284 in the manuscript with revised mark).

9. L. 225, wasn't it said on L. 307-8 that Hoc ab did not neutralize the phage infectivity in vivo?

Fig. 5 b,c and 5 f,g demonstrate only a less than 1-log reduction of the orally applied ETEC. Can they exclude that this small reduction is mediated by the residual luminal phage in the mouse gut? Fig. 5 d,e and 5 h,I lack the phage alone control which would allow such an assessment (distinction of luminal vs. mucus-adherent phage would here be important), particularly since a high dose of 10×10^9 phage was given to a mouse, 10-times more than the challenging ETEC dose.

Reply: Hoc antibody was used to block the Hoc protein in the head of the phage. However, phage infectivity is primarily determined by the tail protein, rendering the blocking of the Hoc protein inconsequential to the phage infectivity. In the manuscript, we demonstrated that Hoc antibody-coated phage had a similar ability to lyse ETEC as non-antibody-coated phage (Supplementary Fig. 3c in lines 242-244 in the manuscript with revised mark).

Supplementary Fig. 3c in the revised manuscript

About the Fig.5b, c and 5f, g in the original manuscript, to exclude that this small reduction is mediated by the residual luminal phage in the mouse gut, we conducted an additional experiment. In this experiment, mice were dissected, and luminal contents and mucus layer of the caecum and colon were collected separately. The data revealed a decrease in the number of ETEC in the phage-pre-treated group compared to the Hoc antibody blocked group, both in the lumen and the mucus layer of the caecum and colon (Fig. 5b - 5e in lines 400-404 of the manuscript with revised mark). The phage number in the mucus layer and lumen of caecum was also significantly decreased after Hoc antibody blocking (Fig. 5f, 5h in lines 404-407 of the manuscript with revised mark). Surprisingly, the number of phage adhering to the mucus layer (Fig.5g in lines 407-411 of the revised manuscript) or the luminal contents (Fig. 5i in lines 407-411 of the manuscript with revised mark) in the colon demonstrated similarity between the Hoc antibody blocked and non-blocked

groups, suggesting that phage replication still occurred in the antibody pretreatment groups, which would have negated the initial impacts of Hoc antibody-blocking. Previous study (doi: 10.1128/mBio.03474-20) has shown that luminal contents contain mucus, which we also observed mucus in the lumen of colon by IFA (Supplementary Fig. 3n in lines 414-417 of the manuscript with revised mark), so that the number of phage decreased both in mucus layer and lumen after NAC or antibody pre-treatment.

We next asked whether the mucus-adherent phage ϕ PNJ-6 could protect the murine intestine from STEC 029 infection. Consistent with the previous results, the intestinal bactericidal capacity of phage ϕ PNJ-6 on STEC 029 was significantly reduced when the Hoc protein was blocked, both in the mucus layer (Fig. 5j, 5k in lines 425-427 of the manuscript with revised mark) and in the luminal contents of cecum and colon (Fig. 5l, 5m in lines 427-428 of the manuscript with revised mark). Correlating with this we observed a significant increase in the number phages between the antibody-blocked and non-blocked groups within mucus layer of the cecum and colon (Fig. 5n, 5o in lines 428-431 of the manuscript with revised mark), with a reduced yet non-significant difference in luminal contents (Fig. 5p, 5q in lines 431-432 of the manuscript with revised mark).

About the Fig. 5 d,e and 5 h,i in the original manuscript, as described above, we have added the phage alone control group in the revised manuscript according to the reviewer's comments.

Fig. 5 in the revised manuscript.

10. L. 392-395: extension of the results on phage therapy is an over-interpretation of the modest less than 1-log ETEC reduction in mice (and the in vitro systems) and lack of attenuation for diarrheal disease in mice (not a model for ETEC diarrhea compared to infant rabbits or pigs).

Reply: We have rewritten this concluding statement and scaled back our phage therapy claims (lines 520-529 in the manuscript with revised mark).

Reviewer #2 (Remarks to the Author):

The authors have conducted a comprehensive study using an *in vitro* cell culture model, an intestinal microfluidic model, and an animal model to uncover the interaction of an *E. coli* phage that binds mucin through its Hoc protein. This interaction leads to increased mucus production, enhanced adhesion in the gut, and ultimately impedes the colonization of *E. coli*, which is an intriguing piece of research. While the manuscript presents valuable experimental work and conclusions, there are a few areas that could be refined:

1. In lines 123-124, the authors mention that the phage pre-treated group in the gut-on-a-chip model can survive normally for up to 24 hours, whereas in the *in vitro* model, almost all of them die after phage pre-treatment. This discrepancy raises questions. Is the difference in medium flow in the gut-on-a-chip influencing the outcome?

Reply: We would like to express our sincere appreciation for your invaluable comments, which have significantly contributed to the improvement of our experiments.

In these experiments, the microbial growth within the supernatant (ie., tissue culture media) rapidly depletes the nutrients available to the cells, resulting in cell death and higher bacterial growth at later time points (e.g., 12 & 24hrs). The gut-on-a-chip model largely overcomes these issues with the use of a pump system that continually washes away microbes in the supernatant and provides replenished nutrients to the system and cell layer. We have included an additional description of the discrepancy between the *in vitro* and gut-on-a-chip systems (see lines 277-284 in the manuscript with revised mark). In summary, this discrepancy is driven by flow fluid and its impact on the microbial community.

2. Additionally, at the end of this paragraph (lines 125-127), it's suggested that the primary reason for the reduced bacterial number is the phage's pre-lysis of the bacteria. However, the text initially discusses the interaction between host cells and bacteriophages. It would be helpful to clarify the basis for this sequence.

Reply: We have revised this sentence in the revised manuscript (in lines 146-148 of the manuscript with revised mark).

3. Figure 1 requires image quality improvement. Fig. 1c lacks clarity, making it hard to discern the scale. Fig. 1i has non-uniform image sizes, and the indicator color for *K. oxytoca* is difficult to distinguish. Consider splitting the channels or using more distinct colors.

Reply: In response to your suggestion regarding the images, we have removed the scale and enhance their clarity (Fig. 1c in the revised manuscript). Specifically, we have enlarged the pictures of *K. oxytoca* to ensure easier differentiation and better visual representation (Fig. 1i in the revised manuscript).

Fig. 1c in the revised manuscript

Fig. 1i in the revised manuscript

4. In lines 165-171, the authors mention contradictory conclusions between the intestinal chip and the *in vitro* model. The purpose of this statement should be clarified. Does it imply that the intestinal microfluidic system better mimics the real situation? Although the advantages of the gut-on-a-chip system are outlined in line 200, it would be beneficial to provide a detailed explanation earlier.

Reply: Yes, we intend to imply that the intestinal microfluidic system better mimics the real situation. In these experiments, the microbial growth within the supernatant (ie., tissue culture media) rapidly depletes the nutrients available to the cells, resulting in cell death and higher bacterial growth at later time points (e.g., 12 & 24hrs). The gut-on-a-chip model largely overcomes these issues with the use of a pump system that continually washes away microbes in the supernatant and provides replenished nutrients to the system and cell layer. We have included an additional description of the discrepancy between the *in vitro* and gut-on-a-chip systems (see lines 277-284 in the manuscript with revised mark).

5. Additionally, the results in Figure 1 obtained by the two methods are consistent, but in Figure 2, they are contradictory. The significance of using two models may need further clarification.

Reply: There are two groups in Figure 1, one is pre-treated with phage, the other is non-pre-treated

with phage, so we cannot compare the number of adhered phage in two groups, thus there is no contradictory results in the number of adherent phage. But in Figure 2, both of two groups pre-treated with phage, so we can compare the number of adhered phage in two groups.

To further explain the contradictory conclusions on phage number, we conducted two additional experiments, which is to investigate the phage affinity for mucus without bacterial interference.

The first experiment is to investigate if mucus effect the adherence of phage. We added NAC to HT-29 cells to remove the mucus and incubated them with ϕ PNJ-6 for 30 min. After incubation, we washed the cells with PBS to remove any unattached phages, and subsequently assessed the phage count. Our findings revealed a lower phage titer in NAC pre-treated group compared to the non-NAC pre-treatment group both in *in vitro* (Supplementary Fig. 2k in lines 202-208 of the manuscript with revised mark) and gut-on-a-chip model (Supplementary Fig. 2l in lines 210-213 of the manuscript with revised mark), suggesting the absence of mucus affects phage adherence.

The second experiment is to investigate if blocking Hoc will affect the phage affinity to mucus. We added ϕ PNJ-6 that had been blocked with Hoc antibodies to HT-29 cells and incubated them for 30 min. After two washes to remove any unattached phages, we determined the phage titer. The results revealed a reduction in phage titers compared to the uncoated antibody group both in *in vitro* (Supplementary Fig. 3f in lines 254-259 of the manuscript with revised mark) and gut-on-a-chip model (Supplementary Fig.3g in lines 254-259 of the manuscript with revised mark), further supporting the promoted effect of Hoc on phage attachment.

Based on these results, we propose that in the absence of bacteria in the *in vitro* model, the phage count is influenced by NAC pretreatment or antibody coating. However, in the presence of ETEC, ϕ PNJ-6 replicates alongside its host resulting in the production of a large number of phages. This results in a higher number of phage within the experimental systems, regardless of the presence or absence of mucin and Hoc.

Supplementary Fig. 2k, 2l in the revised manuscript

Supplementary Fig. 3f, 3g in the revised manuscript

6. The article contains some errors in the text, and it's recommended to conduct a thorough proofreading, including reviewing the figure legends.

Reply: We sincerely appreciate your valuable advice. This manuscript has been carefully and thoroughly edited by a native speaker.

7. The text would benefit from further refinement and elaboration for clarity and coherence.

These improvements will enhance the quality and readability of the manuscript.

Reply: Thank you for your valuable feedback on our manuscript. We have thoroughly refined the manuscript according to your comments.

Reviewer #3 (Remarks to the Author):

The manuscript by Wu et al. delves into the utilization of a novel mucus-adherent phage, øPNJ-6, in the context of phage therapy for gastrointestinal infections. Through a combination of conventional *in vitro* models, a microfluidic gut model, and *in vivo* murine experiments, the research underscores the remarkable capacity of øPNJ-6 to enhance gastrointestinal persistence and exert antimicrobial effects. It accomplishes this by binding to fucose residues in the MUC2 glycoprotein, which, in turn, stimulates increased production of intestinal mucus. Ultimately, this investigation expands the application of the BAM model to phage therapy, confirming its *in vivo* effectiveness and offering valuable insights for future approaches to intestinal phage. Given the current advances of the organ-on-chip systems and the ever-growing interest in the human intestinal microphysiological systems, as well as the potential of phages to treat diseases associated with gut (microbiota), the manuscript is very timely and would likely attract broad interest. However, I have some concerns regarding the model and reported results as detailed below:

1. - ETEC colonization: The authors introduced the ETEC in the dynamic phase along with the flow of the culture medium through the pump, which might reduce the adherence of the pathogens. Instead, the authors should let the ETEC sit in the static phase for a couple of hours and then introduce the flow. This technique has been reported previously for the inoculation of pathogens on a chip (DOI: <https://doi.org/10.1186/s40168-019-0650-5>). The same argument is valid for adding *E. coli* and phages on the chip. Part of the reason for the better results on the chip most likely comes from the fact that the bacteria do not have time to interact with the host due to the approach used by the authors. In contrast, in the *in vitro* setting, they have ample time to adhere and colonize. Therefore, this may not be a fair comparison between the groups.

Reply: We would like to express our sincere appreciation for your invaluable comments, which have significantly contributed to the improvement of our experiments.

We performed an additional experiment according to your comments. In the *in vitro* model, HT-29 cells were pre-exposed to phage for a 30-min incubation period. To ensure consistency between the *in vitro* and gut-on-a-chip models, we implemented a 30-min pause in the pump after introducing øPNJ-6 and ETEC to the gut-on-a-chip.

The data obtained from these experiments revealed that the cell viability in the phage pre-treatment groups in the pump-paused solution exhibited an approximate 20% decrease in comparison to the non-paused phage group. Nonetheless, it remained significant higher cell viability than that observed in the *in vitro* model (Supplementary Fig.3j in lines 267-275 of the manuscript with revised mark). Furthermore, the free phage still showed significant advantage over the Hoc antibody blocked phage or NAC pre-treated group in this pump-paused model (Supplementary Fig.3h-3i in lines 267-272 of the manuscript with revised mark).

Supplementary Fig. 3h-3j in the revised manuscript

2. Challenges with the HT-29 model: Mucus secretion by HT-29 cells is not substantial, and the immunofluorescence (IF) and Western blot images for MUC2 are not satisfactory. The IF images have a lot of background noise (Fig 3g). Additionally, it is very unlikely that these cells secrete a significant amount of mucus during the relatively short gut chip culture period used in this study. The authors should consider staining the mucus with other reagents such as Wheat Germ Agglutinin or Alcian Blue. I recommend using primary or iPSC-derived intestinal organoids as a better model. These models not only secrete more physiologically-relevant mucus but also provide a better mimicry of the human response.

Reply: In our revised study, HT-29 cells were cultured for a duration of 60 h. Subsequently, the Hoc protein was introduced, followed by a 2-h incubation period. To assess the impact on mucin production, the cells were fixed using Carnoy and stained with Alcian Blue. Microscopy was employed to observe the stained cells, and the mean fluorescence intensity (MFI) of mucin in the cells was determined using Image J. The microscopy images clearly demonstrated that the MFI was higher in the Hoc pre-treatment group compared to the non-Hoc groups (Fig. 3g in lines 329-333 of the manuscript with revised mark).

Fig. 3g in the revised manuscript

About mucus secreted in the gut-on-a-chip, it can be obviously observed in Fig.2p (see lines 261-266) in the revised manuscript.

Additionally, we repeated the Western blot to ensure accurate and reliable results (Fig. 3e in the revised manuscript).

Regarding the use of primary or iPSC-derived intestinal organoids, we acknowledge their superiority in mimicking human responses. Regrettably, due to time constraints, we were unable to repeat all the experiments in a new system. However, it is worth nothing that the gut-on-a-chip has already been utilized in pertinent studies, published in PNAS (doi: 10.1073/pnas.1508355112; doi: 10.1073/pnas.2116197119). These studies have garnered recognition from peers, affirming the successful simulation of the intestinal environment to a certain degree using the chip.

3. Measuring barrier function: Authors should measure the barrier function of the intestinal epithelium in the presence of ETEC and ϕ PNJ-6. This would provide a better understanding of the integrity of the epithelium and host-pathogen interactions.

Reply: In response to your query regarding the measurement of the tight junction of the gut barrier in the presence of ϕ PNJ-6 and ETEC, we employed an IFA to assess the expression of E-cadherin. Our findings unequivocally demonstrate that the barrier function of the intestinal epithelium remains intact and robust (Supplementary Fig. 3m in lines 412-414 of the manuscript with revised mark).

Supplementary Fig. 3m in the revised manuscript

4. Gut-on-a-chip images are missing: While the authors have found different results on the Gut Chip platform, all the IF images in the manuscript focus on in vitro static culture. It is essential to include IF images for the Gut Chip to provide a complete picture.

Reply: In response to your insightful comments, we conducted an IFA on the gut-on-a-chip to further investigate the specific aspects you raised. The obtained images indicated that a co-localization between ϕ PNJ-6 and MUC2 within the gut-on-a-chip platform. It was observed that MUC2 levels decreased upon NAC-pre-treatment, leading a subsequent reduction in ϕ PNJ-6. Furthermore, the co-localization was diminished when the antibody against Hoc was used to block its activity (Fig.2p in lines 261-266 of the revised manuscript).

Fig. 2p in the revised manuscript

5. Species-specific pathogenicity of ETEC/E. Coli: It's worth noting that there are species-specific differences in the pathogenicity of ETEC/E. Coli. The real strength of the organ-on-chip system is its potential to offer a more realistic platform for human studies compared to many mouse models. The conclusions drawn about the effect of phage therapy on human E. coli strains and the role of Hoc in phage attachment have primarily been made using animal models. The authors have an opportunity to validate these findings using their Gut Chip system. Many of the observations from mice models may not necessarily translate to human patients, and the human Gut Chip platform could be instrumental in addressing this gap.

Reply: In response to your suggestions, we conducted an experiment involving human isolated strain STEC 029 within the gut-on-a-chip system. During this experiment, the pump was temporarily halted upon after the introduction of the phage or bacteria, as previously described. Our findings revealed ϕ PNJ-6 exhibited a higher concentration (Fig. 5r) within the mucosa, compared to Hoc antibody blocked and NAC pretreatment group, and thus lead to a significant reduction in the load number of STEC 029 (Fig. 5s), accompanied by an increase in HT-29 cell viability (Fig. 5t). Taken together, these results demonstrate that the BAM model is applicable for the targeted reduction of ETEC and STEC originating from both animals and humans within the gastrointestinal mucosa. This was mediated by Hoc adherence to fucosylated mucin glycans, allowing ϕ PNJ-6 to occupy a mucosal niche, increase the production of intestinal MUC2, and subsequently prevent pathogen invasion (lines 433-444 in the manuscript with revised mark).

Fig. 5r -5t in the revised manuscript

REVIEWER COMMENTS

Reviewer #1 (Remarks to the Author):

The authors carefully revised their manuscript according to the comments of the three reviewers making an initially good manuscript even a better report.

Reviewer 1 has only minor comments.

For the rebuttal letter:

there are inherent limitations with continuing....what limitations exactly?

T4-phage, as it is largely unable to infect these clinically relevant ETEC and STEC strains...seems to the reviewer a post-hoc argument after working with another phage since T4 infects also some ETEC and EPEC strains, which can be used in the mouse in vivo model which does not induce diarrhea anyway Fig. 2h ~ 2iwhat explains the difference between caecum and colon mucus binding?

for a duration of 60 h to establish an appropriate cellular environment.....HT29 cells need more than 60 h for differentiation in cell culture; which might explain the somewhat weak mucus expression suggesting that phage replication still occurred in the antibody pretreatment groups.... What is the evidence that phage replication occurred in vivo? Difficult to demonstrate in view of the high phage inoculation dose.

Main document:

I. 22, 40: viruses are not microorganisms sensu strictu, but cellular parasites

I.43-45: sentence "promise...unclear" logical contradiction, reformulate

I.60-61 unclear, reformulate

I.69: commensal or any gut bacterium, not just symbiotic ones

I.73 phage species... really, not phage types

I.111: enhances phage replication... erroneous interpretation of the presented figure, the peak titer is the same, mucins seem to stabilize phage against decay

I.118: measurable...yes, but weak; 60h preincubation might be too short

I. 190-1: does this observation not contradict the main hypothesis of the manuscript?

I. 207: significant, but weak

I.214: impeded better: reduced

L. 2223-4: 2-to 3-fold... this is a weak effect

L. 248-250: this remains a disturbing discrepancy

I. 334: what means tripartite here?

I.376: preferentially ...skip , you mean pre-exposed/ pre-treated?

I.410: phage replication....a helpful (even plausible) ad hoc hypothesis but based on what evidence?

I. 448: a truism, skip

I.449: value...really? Quote evidence

I. 450-3: vague propose to skip the first paragraph

I. 453-7: another circular argument

I. 467 typo STEC

I. 469-70: this is an overinterpretation of the experiments, skip

I. 471 trivial, skip; what else remains: lipids?

I. 511-9: the extrapolation to IBD is highly speculative and better skipped

Reviewer #3 (Remarks to the Author):

Authors have addressed my questions.

Reviewer #1 :

The authors carefully revised their manuscript according to the comments of the three reviewers making an initially good manuscript even a better report.

Reviewer 1 has only minor comments.

For the rebuttal letter:

There are inherent limitations with continuing...what limitations exactly?

T4-phage, as it is largely unable to infect these clinically relevant ETEC and STEC strains...seems to the reviewer a post-hoc argument after working with another phage since T4 infects also some ETEC and EPEC strains, which can be used in the mouse in vivo model which does not induce diarrhea anyway.

Reply: We greatly appreciate your questions and comments! Regarding the use of T4 phage as a model, the limitations we were referring to are not purely technical in nature. Regarding the non-technical limitations, we felt that continuing to focus on T4 phage would limit the broader application of the BAM model and our potential to uncover novel biology. As such, we made the decision to expand beyond the already well-characterized T4 phage system and move into a novel phage-host pair. This led us to our decision to focus this work and our characterization efforts on the novel phage ϕ PNJ-6, which infects the gastrointestinal pathogen ETEC. From this work, we have demonstrated the ϕ PNJ-6 protects epithelial cells from bacterial infection and adheres to fucosylated mucin glycoproteins via the externally displayed Hoc protein, all of which are in line with the previously characterized T4 phage model. In addition, we have found that ϕ PNJ-6 and its Hoc protein led to an upregulation in MUC2 expression, which had not previously been described, and identified a unique glycan binding pocket within domain 1 of the ϕ PNJ-6 Hoc protein, which is functionally different to the described domain 3 binding pocket in the T4 phage model.

As the reviewer has highlighted, T4 phage might infect some ETEC and EPEC strains. While it is possible to apply T4 phage to our in vivo model, the reality is that this would require significant experimental testing and validation, and the time and costs required to do this are outside of the scope of our current manuscript.

In summary, while this study does suffer from the limitation of the well-defined T4 phage and its Δ Hoc mutants, we believe that the discovery and characterization of the ϕ PNJ-6 phage provides novel insights and continues to build upon this research area.

Fig. 2h ~ 2i ...what explains the difference between caecum and colon mucus binding?

Reply: As for Fig 2h ~2i, the explanation provided is that the mucus layer in the colon is constantly renewed by goblet cells, causing metabolized mucus-containing phages to be excreted through intestinal peristalsis. This process may result in a lower number of phages adhering to the colon mucus compared to the cecum mucus. Additionally, the pouch-like structure of the cecum may facilitate phage retention, leading to a greater difference in phage numbers between the NAC-treated and untreated groups in the colon than in the cecum. These points have been elaborated on in lines 491-498 of the manuscript with revised marks.

For a duration of 60 h to establish an appropriate cellular environment...HT29 cells need more than 60 h for differentiation in cell culture; which might explain the somewhat weak mucus expression

Reply: We sincerely appreciate your valuable comments and concede this point as a limitation, which has been to the revised manuscript, please see lines 499-501.

Suggesting that phage replication still occurred in the antibody pretreatment groups... What is the evidence that phage replication occurred in vivo? Difficult to demonstrate in view of the high phage inoculation dose.

Reply: It is very hard to quantify whether phage replication occurred in vivo or not. However, based on our previous experimental data, when phage blocked with Hoc antibody was administered to mice by gavage in the absence of host bacteria, phage could not be detected in the cecum and colon at 18 h and 12 h (Fig. 2n, 2o), respectively. However, when host bacteria were present, a significant number of phages could still be detected in the colon and cecum 18 h after gavage with blocked phage (Fig. 5f, 5g, 5h, 5i). This may be due to phage replication in the presence of host bacteria. As there is no direct evidence, we will change this conclusion to a hypothesis. Please see lines 386-387 in the revised manuscript. Besides, relevant content has been added to the discussion in lines 475-479 of the manuscript with revised marks.

Main document:

l. 22, 40: viruses are not microorganisms sensu strictu, but cellular parasites

Reply: Thank you for your accurate and rigorous advice. We have revised based on your suggestion in lines 21, and 38 of the manuscript with revised marks.

l.43-45: sentence “promise...unclear” logical contradiction, reformulate

Reply: We have rewritten this sentence in lines 41-44 of the revised manuscript.

l.60-61 unclear, reformulate

Reply: We have reformulated this sentence in lines 55-58 of the revised manuscript.

l.69: commensal or any gut bacterium, not just symbiotic ones

Reply: We have removed it in line 68 of the revised manuscript.

l.73 phage species... really, not phage types

Reply: We have modified it according to your suggestion in line 71.

l.111: enhances phage replication... erroneous interpretation of the presented figure, the peak titer is the same, mucins seem to stabilize phage against decay

Reply: We have made changes based on your comments, please see lines 106-107.

l.118: measurable...yes, but weak; 60h preincubation might be too short

Reply: We have modified it according to your comment in line 114 of the revised manuscript with marked, and we have pointed out this is a limitation of our manuscript, please see lines 499-501.

l. 190-1: does this observation not contradict the main hypothesis of the manuscript?

Reply: As for Fig. 2a, 2b, regarding the contradictory results observed across the *in vitro* and gut-on-a-chip models, we have carefully considered the possibility that the presence of ETEC may be responsible for the divergent outcomes in the two experiments. Under *in vitro* conditions where there is no fluid flow, stagnant conditions facilitate the propagation of both ϕ PNJ-6 and ETEC in the supernatant where the effect of phage-mucus interaction is marginal. Comparatively, within the gut-on-a-chip microfluidic device, continual fluid flow across the mucosal surface facilitates the removal of microorganisms in the supernatant and amplifies any mucosal interactions. Based on this speculation, we conducted additional experiments of bacteria-free. We found that in the absence of bacteria in both *in vitro* and gut-on-a-chip models, phage titer decreased significantly after NAC pre-treatment (Supplementary Fig. 2k, 2l).

In vitro, microbial growth within the supernatant (ie., tissue culture media) rapidly depletes the nutrients available to the cells, resulting in cell death and higher bacterial growth. The gut-on-a-chip model largely overcomes these problems with the use of a

pump system that continually washes away microbes in the supernatant and provides replenished nutrients to the system and cell layer, which more closely resembles the in vivo situation. This result also illustrates the superiority of the gut-on-a-chip.

l. 207: significant, but weak

Reply: As for Supplementary Fig. 2k, the change in the number of phage in the NAC-pretreated group was weak, probably due to the fact that NAC did not completely remove the mucus, but merely caused it to decrease. The use of the *muc2* knockout cell line may further compensate for the shortcomings of this test. We have pointed out this as a limitation in lines 501-505 of the manuscript with revised marks.

l.214: impeded better: reduced

Reply: We have revised it according to your suggestion in line 200.

L. 2223-4: 2-to 3-fold... this is a weak effect

Reply: As for Fig 2f, 2j, as mentioned above, the change in the number of phages in the NAC-pretreated group was weak, probably due to the fact that NAC did not completely remove the mucus, but merely caused it to decrease. The use of *muc2* knockout mice may further compensate for the shortcomings of this experiment. We have pointed out that this is a limitation of our manuscript, please see lines 501-505.

L. 248-250: this remains a disturbing discrepancy

Reply: As for Fig 2j, 2k, as mentioned above, we speculated that the presence of ETEC may be responsible for the divergent outcomes. From the results of the bacteria-free, we found that phage titer decreased significantly after Hoc antibody coated in vitro (Supplementary Fig. 3f). Regardless of the presence of bacteria, the consistency between chip experiments and in vivo experiments demonstrates the superiority of the chip model. We have explained this content in lines 256-265 of the manuscript with revised marks.

l. 334: what means tripartite here?

Reply: We have modified this sentence in lines 313-314.

l.376: preferentially ...skip , you mean pre-exposed/ pre-treated?

Reply: We mean pre-exposed, and we have modified it in line 355.

l.410: phage replication....a helpful (even plausible) ad hoc hypothesis but based on what evidence?

Reply: It is very hard to quantify whether phage replication occurred in vivo or not. However, based on our previous experimental data, when phage blocked with Hoc antibody was administered to mice by gavage in the absence of host bacteria, phage could not be detected in the cecum and colon at 18 h and 12 h (Fig. 2n, 2o), respectively. However, when host bacteria were present, a significant number of phages could still be detected in the colon and cecum 18 h after gavage with blocked phage (Fig. 5f, 5g, 5h, 5i). This may be due to phage replication in the presence of host bacteria. As there is no direct evidence, we will change this conclusion to a hypothesis in lines 386-387. These points also have been elaborated on in lines 475-479 of the revised manuscript with marks.

l. 448 : a truism, skip

Reply: We moved it according to your suggestion in lines 419-423 of the manuscript with revised marks.

l.449: value...really? Quote evidence

Reply: We have moved it according to your suggestion in lines 419-423.

l. 450-3: vague propose to skip the first paragraph

Reply: We removed it based on your comments, please see lines 419-423.

l. 453-7: another circular argument

Reply: We apologize for the redundancy of the wording, and we have made changes to make the wording more concise based on your comments, please see lines 424-427 of the manuscript with revised marks.

l. 467 typo STEC

Reply: We have revised it, please see line 437.

l. 469-70: this is an overinterpretation of the experiments, skip

Reply: We have moved any overstated conclusions and bold extrapolations in lines 438-440 and listed the limitations of this study in lines 499-506.

l. 471 trivial, skip; what else remains: lipids?

Reply: We have removed this sentence in line 441.

l. 511-9: the extrapolation to IBD is highly speculative and better skipped

Reply: Thanks again for your rigorous and objective advice and we have removed this section in response to your comment, please see lines 483-490.